# Positive feedbacks and alternative stable states in forest leaf types

Yibiao Zou [1] ✉, Constantin M. Zohner [1], Colin Averill [1], Haozhi Ma [1], Julian Merder [2], Miguel Berdugo [1], Lalasia Bialic-Murphy[1], Lidong Mo [1], Philipp Brun [3], Niklaus E. Zimmermann [3], Jingjing Liang [4], Sergio de-Miguel [5,6], Gert-Jan Nabuurs [7], Peter B. Reich [8,9,10], Ulo Niinements[11], Jonas Dahlgren[12], Gerald Kändler[13], Sophia Ratcliffe[14], Paloma Ruiz-Benito[15], Miguel Angel de Zavala [15], GFBI consortium* & Thomas W. Crowther [1]

The emergence of alternative stable states in forest systems has significant implications for the functioning and structure of the terrestrial biosphere, yet empirical evidence remains scarce. Here, we combine global forest biodiversity observations and simulations to test for alternative stable states in the presence of evergreen and deciduous forest types. We reveal a bimodal distribution of forest leaf types across temperate regions of the Northern Hemisphere that cannot be explained by the environment alone, suggesting signatures of alternative forest states. Moreover, we empirically demonstrate the existence of positive feedbacks in tree growth, recruitment and mortality, with trees having 4–43% higher growth rates, 14–17% higher survival rates and 4–7 times higher recruitment rates when they are surrounded by trees of their own leaf type. Simulations show that the observed positive feedbacks are necessary and sufficient to generate alternative forest states, which also lead to dependency on history (hysteresis) during ecosystem transition from evergreen to deciduous forests and vice versa. We identify hotspots of bistable forest types in evergreen-deciduous ecotones, which are likely driven by soil-related positive feedbacks. These findings are integral to predicting the distribution of forest biomes, and aid to our understanding of biodiversity, carbon turnover, and terrestrial climate feedbacks.

Alternative stable states exist in ecological, climatic, and social systems[1–3]. In such systems, feedbacks maintain the state of the system unless gradual forcing or perturbations become too large and cause abrupt, critical transitions between stable states[2]. An important example of alternative biome states is the forest *versus* savanna distinction, whereby fire feedbacks play a key role in maintaining one or the other state[4,5]. Yet, it remains unclear whether different tree functional groups form alternative stable states within forest systems, and

what feedbacks might drive them, limiting our capacity to predict state changes that affect terrestrial carbon turnover, water dynamics and nutrient cycling[6].

Forests are either deciduous or evergreen or a mix of the two[7], and the distribution of these forest leaf types underlies dynamic global vegetation models[8–10]. Deciduous trees that shed all leaves during unfavorable periods differ from evergreen trees in a variety of ecologically- and climate-relevant leaf traits, such as life span, nutrient

A full list of affiliations appears at the end of the paper. *A list of authors and their affiliations appears at the end of the paper.

✉e-mail: yibiao.zou@usys.ethz.ch

concentration, and photosynthesis, respiration, and decomposition rates[7,11,12]. Whether evergreen and deciduous forests form alternative stable states or not thus fundamentally determines the potential abundance of forest leaf types[13,14], affecting global conservation and restoration efforts, the structure and function of forest ecosystems[12,15,16] and the global climate system[9,17].

The global distribution of leaf phenology strategies (evergreen *versus* deciduous) is strongly linked to environmental conditions, with evergreen species being most abundant in warm, a-seasonal, and humid regions (broadleaf evergreen) or cold and nutrient-poor regions (needleleaf evergreen)[7,18,19]. However, evidence suggests that biological feedbacks within forest stands may also drive leaf phenology strategies, whereby the dominant type in a community may favor the establishment and survival of its own type (the con-phenological feedback)[10,14,20]. For example, many evergreen trees (especially conifers) have an advantage over deciduous species in nutrient-poor and acidic soils[7,20]. High concentrations of tannins and phenols as well as low N concentration in evergreen leaves decrease the soil pH and rates of leaf decomposition, further limiting soil fertility and, in turn, favoring the dominance of evergreen species[7,21,22]. Similarly, deciduous trees may also favor their own phenology type by shedding less tannic, nutrient-rich leaves that can quickly be decomposed[23]. Furthermore, the dominance of either leaf type can lead to an accumulation of 'con-phenological' seeds and seedlings, which may strengthen the positive feedback[24]. If these positive feedbacks are strong enough, they can generate alternative stable states of evergreen and deciduous forests[25], with strong implications for the resilience of ecosystem structure and functioning[1].

Although positive feedbacks have been proposed by theoretical studies to stabilize alternative stable states of leaf phenology strategies[21,26], empirical evidence for this hypothesis is still scarce. This is partly due to trees' slow growth and turnover rates, which complicate the acquisition of time-series data across multiple generations — essential for testing alternative stable states in forest ecosystems[14]. In addition, multiple competing hypotheses[24,27–29] limit the empirical

testing of alternative stable states. Further, we lack a spatial understanding of where alternative forest phenological states are likely to be present, and what specific factors might drive these biogeographic patterns.

Emerging approaches reveal how different criteria can be used to empirically test their presence[14,30]. For instance, if phenological alternative-stable-states exist, then most forests should be dominated by either evergreen or deciduous trees, with mixed forests being rarer than predicted by chance[31]. Therefore, bimodal patterns of forest types should represent signatures of alternative stable states if the effects of other confounding processes that can also generate bimodality, such as environmental filtering and monoculture plantation[27,28], are controlled for[30]. As such, the presence of alternative stable states can be tested using multiple distinct lines of evidence[30,31]: (i) the frequency distribution of observed leaf phenology strategies is bimodal, and this cannot be explained by environment or management alone. (ii) Demographic positive feedbacks in recruitment, growth and survival lead to the promotion of the same leaf phenology type. (iii) The observed feedback is strong enough to generate and maintain bimodality through time under environmental heterogeneity and demographic stochasticity. (iv) Dependency on initial conditions (hysteresis) exists during transition from one stable state to another. When these four criteria align, this provides evidence that the bimodal distribution is the result of alternative stable states, stabilized through feedback processes.

We here use a combination of empirical data analysis and data-driven simulations to (a) test these four criteria for the existence of alternative stable states in leaf phenology, and (b) quantify the spatial extent of this phenomenon. We reveal bimodality in the distribution of forest leaf phenology types, even after accounting for environmental filtering and monoculture plantations, and quantify the con-phenological demographic feedbacks underpinning these bimodal patterns. We also show that the observed feedbacks are necessary and sufficient to generate and maintain alternative stable states of leaf phenology, which lead to hysteresis during ecosystem transition. Our

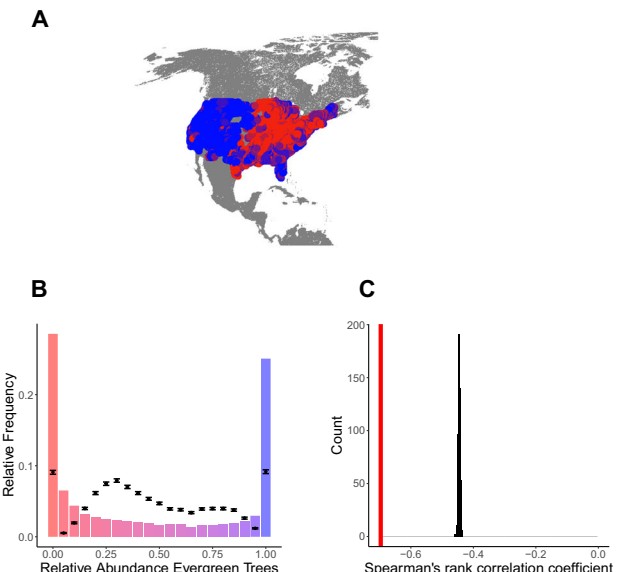

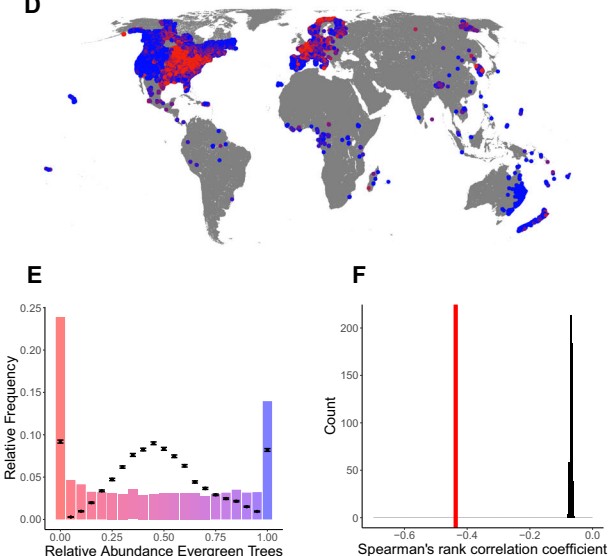

**Fig. 1 | Bimodal patterns of forest types at the continental and global scale.** **A** The spatial distribution of 45,276 forest inventory sites from the FIA was used for the continental analysis. The continuous color scale represents the relative abundance of evergreen trees within a plot (red = 100% deciduous; blue = 100% evergreen). **B** Histogram of the observed plot-level evergreen percentage across the mainland US. The black dots and error bars show the medians and 2.5–97.5% quantiles of the null model predictions driven by environmental filtering (zero adjusted Poisson distribution). **C** Spearman's rank correlation coefficient between

evergreen abundance and deciduous abundance in the observed data (red bar) versus the simulated results of the null model (black histogram). **D** The location of 815,578 forest plots from the global GFBI database. **E** and **F** are the same as panels **B,** and **C**, but for the global data. Hartigan's dip test[52] showed significant multi-modality (here is bimodality) in the observed values in panels **B** (n = 45,276, one-sided *p*-value < 0.001) and **E** (n = 815,578, one-sided *p*-value < 0.001). Source data are provided as a Source Data file.

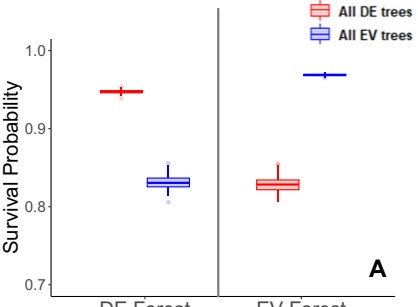
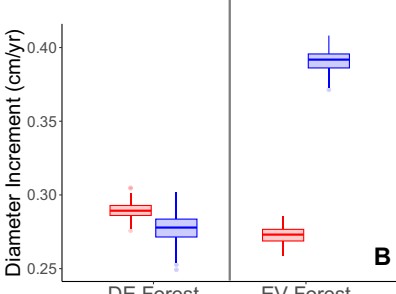
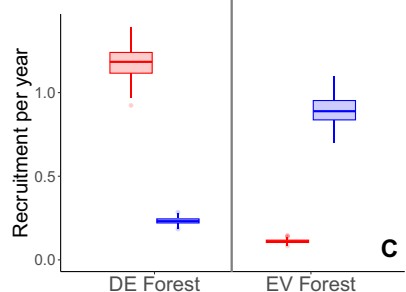

**Fig. 2 | Observed positive feedback in con-phenological demographics in the mainland US. A** Survival probability of an individual deciduous tree (DE, red) or evergreen tree (EV, blue) within a purely evergreen or deciduous forest stand. **B** Individual deciduous or evergreen tree growth (stem diameter increment in cm per year) when the surrounding trees are purely evergreen or deciduous. **C** Recruitment rates of deciduous or evergreen trees in deciduous or evergreen dominated forest plots. All plotted data are 100 samples drawn from the 95% CI of the corresponding full model ($n = 45,276$), controlling for environmental conditions and stand structure. Results are presented as boxplots (medians as centre with 25th and 75th percentile as bounds). The differences between all compared pairs are highly significant (one-sided t-test $p$-value < 0.001, $n = 100$). Source data are provided as a Source Data file.

spatial models further revealed hotspots of alternative stable states in evergreen-deciduous ecotones across temperate regions of the Northern Hemisphere.

## Results

### Bimodal patterns of forest types

To test whether bimodal patterns exist at the continental and global scale and whether environmental filtering is sufficient to explain these patterns, we used forest plot data from the Forest Inventory and Analysis (FIA) Program[32] of the US and the Global Forest Biodiversity Initiative (GFBI)[33]. Species-level leaf phenology classification (evergreen *versus* deciduous) came from the TRY database[34], and we computed the plot-level relative abundance of leaf phenology strategies based on the stem density (number of stems per plot) of each type. To control for the effects of human management, we removed managed plots from the FIA data and monoculture plots (containing <2 species) from the GFBI data. The remaining data show clear bimodality in leaf phenology strategies, both at the continental and global scale (Fig. 1B, E).

To control for the effects of environmental filtering, we fitted generalized additive models (GAMs)[35] to each phenological type. We modeled plot-level evergreen/deciduous stem density as a function of the ten leading environmental principal components, covering impacts of climate, topography and human activity. We then sampled random evergreen and deciduous stem densities for each plot from the distribution expected under plot-specific environmental conditions using the GAMs, and computed the relative evergreen abundance in each plot (Fig. 1B, E). The data-model comparison shows that environmental filtering alone is not sufficient to reproduce the magnitude of bimodality at both the continental and global scale (Fig. 1B, E).

The alternative stable state hypothesis further predicts that the absolute abundances of evergreen and deciduous trees at the plot level are more negatively correlated than what would be expected if only environmental filtering were to drive leaf-type abundances. Indeed, the negative correlation between both leaf strategies was much larger in the observed data than in the ensemble of modeled abundances ($p < 0.001$, Fig. 1C, F), suggesting that factors apart from environmental filtering drive leaf type distributions. This pattern remained similar when correcting for overdispersion and when applying the same analysis to European forest inventory data (Fig. S3). Together, these results show that both environmental variables and overdispersion cannot fully explain leaf-type variation at the continental and global scale, instead pointing toward other mechanisms that drive alternative stable states of forest leaf phenology.

### Strong positive feedbacks in forest con-phenological demography

To explore whether demographic positive feedbacks drive the observed bimodal patterns, we modeled tree mortality, growth, and recruitment using 45,276 repeatedly measured forest inventory plots across the mainland US. We used GAMs to incorporate and control for the influence of environmental covariates, allowing us to separate the influence of ecological feedbacks from potential covariation with the environment. After controlling for environmental covariates, stand conditions, and tree size, the results showed strong positive con-phenological feedbacks, that is, trees perform better when surrounded by their own phenological strategies. In particular, we found that a deciduous tree has a 14 ± 1% (mean ± 95% CI) higher probability of survival than an evergreen tree when both are growing in a forest dominated by deciduous trees, while an evergreen tree has a 17 ± 1% higher survival probability than a deciduous tree when both are growing in an evergreen-dominated forest (Figs. 2A and S3). Similarly, we found that, in evergreen-dominated forests, evergreen trees grew 43 ± 4% faster than deciduous trees, whereas in deciduous forests, deciduous trees grew 4 ± 3% faster than evergreen trees (Figs. 2B and S3). Furthermore, in evergreen forests, recruitment of evergreen trees was much higher than that of deciduous trees and vice versa (Figs. 2C, S3). Species-level analyses confirmed this, showing con-phenological facilitation of recruitment and survival among the 20 most abundant evergreen and deciduous tree species in the US (Fig. S4). Demographic models refit to ecoregions of the eastern US showed robust positive demographic feedbacks at the sub-regional scale (Fig. S5). The same patterns were found for European forest data (Fig. S6). These results support the existence of con-phenological feedback, which may explain bimodality of phenological strategies.

### Effect of demographic feedbacks on forest succession

To understand if the con-phenological demographic feedbacks observed over relatively short time scales have the potential to generate alternative stable states over long timescales, we ran a set of forest demographic simulations. Specifically, we ran two groups of simulations based on empirically fitted GAMs for growth, recruitment, and mortality[30]. The first group of simulations allowed us to test whether this feedback can generate alternative stable states from uniformly mixed forests over long time periods (i.e., over multiple tree generations), and the second group tests whether the feedback can maintain alternative stable states.

The results showed that null model simulations could not generate and maintain a bimodal distribution of evergreen versus

deciduous forests. In contrast, simulations driven by feedback models reproduced the observed bimodal pattern of forest leaf phenology (Fig. 3A–D). We then extended these demographic simulations to the 20 most abundant evergreen and deciduous species in the US to show that the feedback model could maintain bimodal patterns, whereas the null model couldn't (Fig. S10). These findings suggest that positive demographic feedbacks are sufficient to generate and maintain alternative stable states under demographic stochasticity, environmental heterogeneity, and disturbances.

### Hysteresis

The presence of alternative stable states in forest leaf phenology strategies predicts that hysteresis will occur during the transition of phenology strategies along environmental gradients. In temperate regions, mean annual temperature (MAT) has been well documented to drive the transition from evergreen forests (mostly needleleaf) to deciduous forests (mostly broadleaf)[7,36]. To test whether hysteresis occurs during the transition from one dominant phenology type to the other in response to temperature, we ran a third group of simulations implemented along a gradient of MAT (−2 °C to +23 °C, the empirical range of MAT in forests across the mainland US). We split the MAT gradient into 12 sections and initialized 1000 forests plots for each section as either evergreen-dominated (800 purely evergreen plots + 200 purely deciduous plots), or deciduous-dominated (800 purely deciduous plots + 200 purely evergreen plots). We simulated both 'null' (excluding feedback predictors) and 'feedback' scenarios for 2000 years to minimize the influence of demographic lags on the model outcome. The feedback simulations accurately predicted the observed decrease in evergreen dominance with increasing MAT[7,37] and revealed clear signs of hysteresis, with the final leaf phenology type of forest plots under any given MAT strongly depending on the initial phenological status (Fig. 3D). Within each MAT group, the relative abundance of evergreen species was higher under the evergreen-dominated initialization than under the deciduous-dominated initialization. In contrast, null simulations didn't show significant initialization-dependent differences in final abundances, indicating con-phenological feedback as the primary driver of hysteresis (Fig. 3C).

### Spatial extent of alternative stable states in forest systems

Alternative stable states in leaf phenological strategies are region-specific, as environmental filtering may limit certain regions to only one leaf type[7,38]. To generate a spatial understanding of the potential presence of alternative stable states, we developed a random forest model (see workflow in Fig. S8). We partitioned the global forest zones using a 'fishing net' with 10 arc-min (~20 km) grid size. For each cluster, we used forest composition information of all plots to calculate a bimodality index (BI), which helped quantify bimodality in the leaf phenology distribution (Fig. 4A). The BI ranges from −1 to 1, and we empirically derived bimodality cutoffs, with BIs < −0.22 representing deciduous-dominated clusters, BIs > 0.22 representing evergreen-dominated clusters, and BIs of −0.22–0.22 representing bimodal clusters (Fig. 4C).

To extrapolate the BI across the Northern Hemisphere, we trained a random forest model, including 62 environmental predictors covering spatial variation in climate, soil, topography, vegetation and human impact (Fig. 4). The random forest models explained 71% (coefficient of determination based on 10-fold cross validation) of the global variation in the bimodality of leaf phenology. Spatially buffered leave-one-out cross-validation showed that performance of the random forest model remained satisfactory ($R^2 = 0.52$) at a 500 km buffer radius, at which scale no spatial autocorrelation in model residuals was detected anymore (Fig. S12). The maps show that bimodal forests mainly occur in ecotones between evergreen and deciduous forests and at the poleward range limits of forest ecosystems (Fig. 4A, B).

An alternative approach using environmental instead of spatial clusters showed consistent patterns (72% agreement between models, Figs. S8 & S16). The models thus capture phenological bimodality observed in well-studied ecotones like hemlock vs. maple forests in northern Michigan[39], US (predicted BI = −0.11), evergreen-oak vs. deciduous forests in the Sierra Madre[37], Mexico (predicted BI = 0.10) and evergreen conifer vs. deciduous lime forests in southern Sweden[40] (predicted BI = 0.20).

Global variation in leaf phenological bimodality was best explained by climatic variables, such as mean annual temperature and temperature of the coldest quarter, while soil conditions played a subordinate role (Figs. 5A, S23A & S24A). Yet, within bimodal clusters, soil pH was the most important driver of the relative proportion of evergreen trees in a plot, followed by soil nitrogen and annual precipitation (Figs. 5B, S23C & S24C). In both evergreen and deciduous dominated clusters where there are no alternative stable states present, climatic factors were more important than soil factors in explaining relative evergreen abundance (Figs. 5B, S23C & S24C). A principal-component-based analysis showed the same patterns (Fig. S11), supporting that soil conditions, particularly soil pH, play an important role in driving variation of forest phenological composition in bimodal clusters.

## Discussion

The spatial distribution of evergreen and deciduous forests, as well as the factors governing their formation, have long been of great interest to ecologists[7,20]. To test for the existence of alternative stable states in leaf phenological types and explore the involved mechanisms, numerous theoretical studies have simulated positive feedback loops between evergreen conifers, deciduous species, and their physical environment, including soil nutrients, water availability, and climatic conditions[10,18,41]. Building upon this foundational research, we found multiple lines of empirical evidence to suggest that the global distribution of deciduous and evergreen tree species is partially underpinned by ecological feedbacks that drive forest ecosystems towards alternative stable states. The strong bimodality in the distribution of deciduous and evergreen species can be observed even after controlling for environmental filtering and overdispersion (Fig. 1), suggesting that each leaf phenology type favors their own forest type via positive feedback. In addition, we found higher growth, survival and recruitment rates of deciduous trees in deciduous stands than in stands dominated by evergreen species, and vice versa (Fig. 2). Moreover, we were only able to reproduce the bimodal pattern in empirically derived dynamical models when incorporating the feedbacks. These ecological feedbacks are therefore likely to gradually shift the distribution of forest types towards distinct alternative stable states in certain environmental envelopes across the global forest system.

Accounting for such biological feedbacks is particularly relevant for biogeochemical modeling efforts that aim to represent the functional variation in forest types across the globe. For example, to predict the presence of forest functional types, dynamic global vegetation models currently use deterministic climate limits to predefine the area where functional types can establish and then allow functional types that pass the climatic filter to coexist and compete for resources[8,9]. Historic community composition, however, is ignored when predicting the dominant functional type of an area[8,9]. Our models support this approach in that climatic constraints strongly drive the global distribution of forest types. Yet, we also reveal that, even under the same climate conditions, the initial leaf phenology type of an ecosystem can affect its final stable state (Fig. 3E, F), highlighting the importance of including historic ecosystem status in dynamic global vegetation models.

Our data and spatial models indicate hotspots of leaf phenological alternative stable states in ecotones and the poleward range limits of

**Simulation 1: succession with uniform initialization**

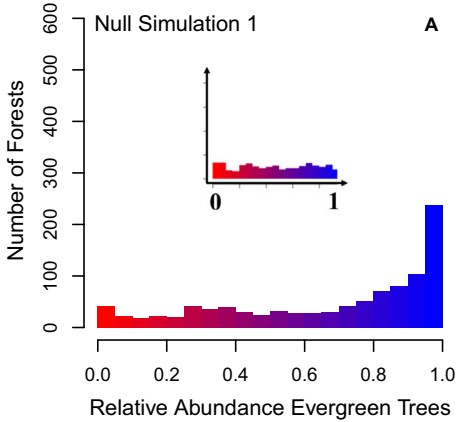
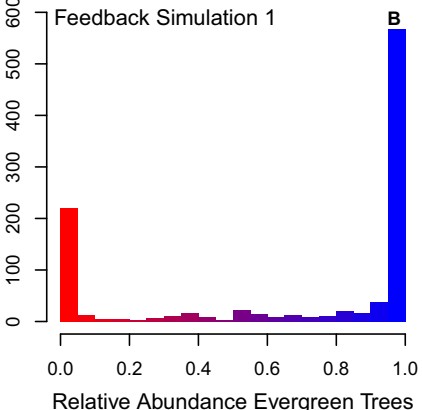

**Simulation 2: succession with bimodal initialization**

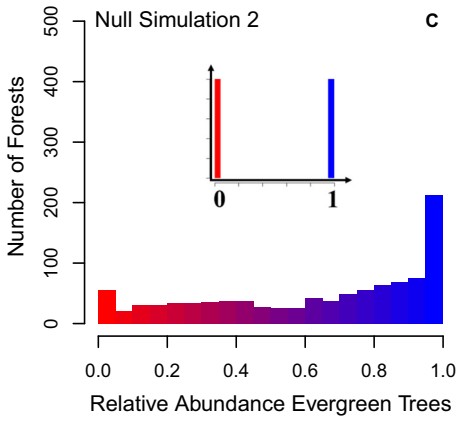
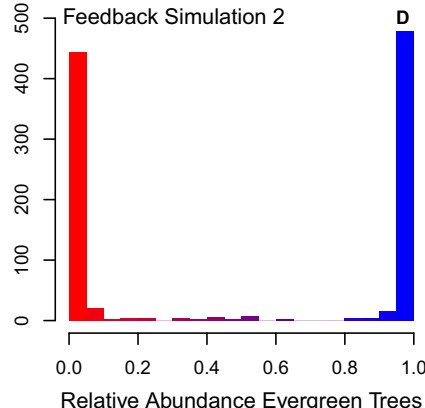

**Simulation 3: hysteresis analysis**

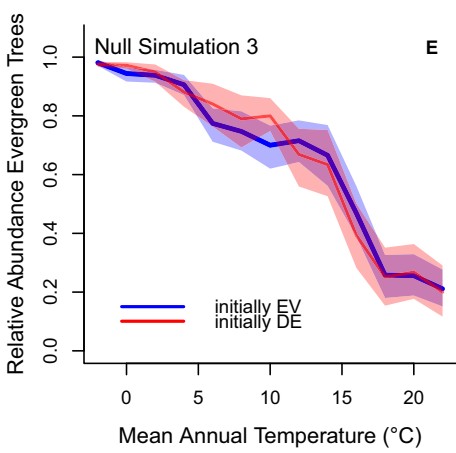
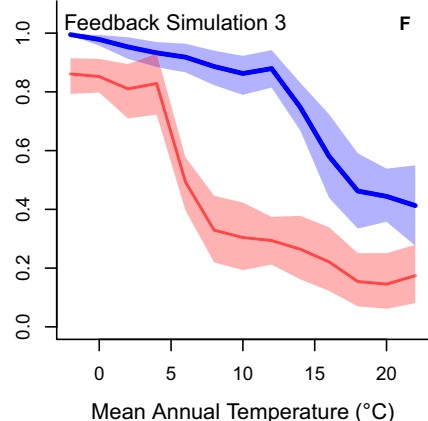

forests (Fig. 4B). Ecotones between evergreen- and deciduous-dominated regions are typically characterized by environmental conditions that allow both leaf phenology strategies to coexist, thus allowing each type to form patches (Fig. 4C middle panel) via soil-related positive feedbacks[5,26,37]. Studies on alternative stable states of forest vs. savanna systems have also shown that bistable regions are located in-between forest- and savanna-dominated areas[4,5]. The causes behind the presence of bimodal clusters at poleward tree range limits, however, remain elusive and warrant further investigation. The dependence of forest composition in these bimodal regions on the

initial ecosystem state cautions against simply using environmental conditions to predict forest types in these areas.

At the global scale, the distribution of evergreen, deciduous, and bistable forests is mainly determined by climatic variables. In contrast, within bistable forest clusters, soil chemical variables best explained variation in leaf phenology composition. This suggests that soil-related positive feedbacks shape the bimodal patterns, supporting our hypothesis that trees facilitate establishment of their own phenology type by regulating soil properties like pH and nutrient content. Trees can modify the soil pH via their litterfall, and a soil pH around 4.5 has

**Fig. 3 | Three pairs of feedback simulations versus null simulations.** Histograms in **A**–**D** represent the relative evergreen abundance within 1000 simulated forest plots after 2000 years. The color scale represents the percentage of evergreen forest within a plot (red, 100% deciduous; blue, 100% evergreen). **A**, Outcome of null demographic simulations with uniform initialization, from growth, recruitment, and mortality models fit without con-phenological feedback predictors. The inset in (A) shows the initial uniform distribution of relative evergreen abundance across the 1000 plots. The uniform distribution shifts to evergreen-dominated forests after 2000 years. **B**, Outcome of demographic simulations with uniform initialization from demographic models fit with con-phenological predictors. Most plots are dominated by either of the two leaf phenology strategies (bimodal distribution). **C** Outcome of null demographic simulations with bimodal initialization, from growth, recruitment, and mortality models fit without con-phenological feedback predictors. The inset in panel C shows the initial bimodal distribution of forest composition. As in panel A, the bimodal distribution shifts to evergreen-dominated forests after 2000 years. **D** Outcome of demographic simulations with bimodal initialization from demographic models fit with con-phenological predictors. Most plots are dominated by either of the two leaf phenology strategies. **E**, **F** Hysteresis simulations along mean annual temperature gradients, where 80% of forest plots (800 of 1000) were initially dominated by evergreen trees (EV, blue line) or deciduous trees (DE, red line). Demographic simulations were run either without feedback predictors (**E**) or with feedback predictors (**F**). Shaded regions represent the 95% CI of the mean relative abundance of evergreen trees across each set of simulations ($n = 1000$). Source data are provided as a Source Data file.

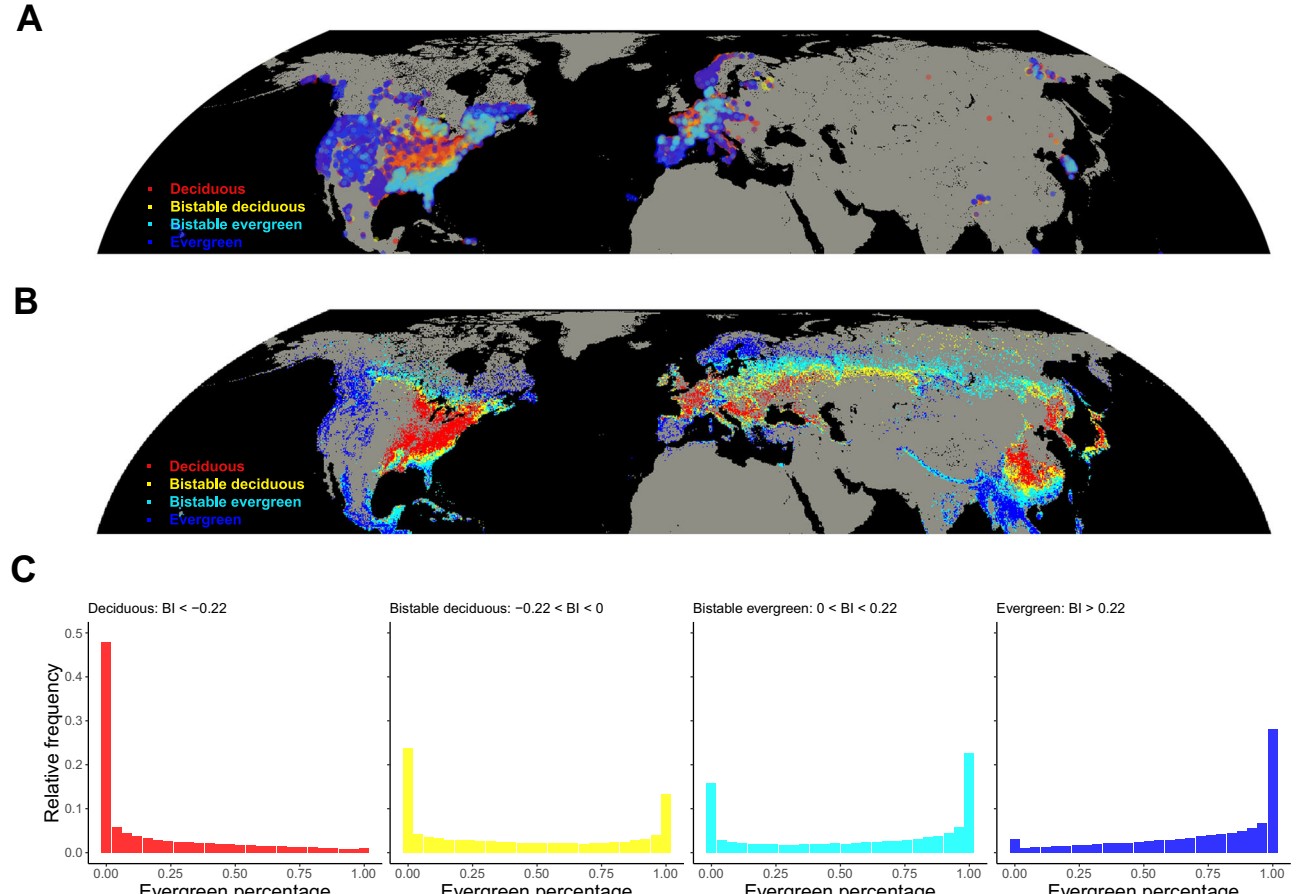

**Fig. 4 | Hemisphere-wide patterns of forest types. A** Empirical map of bimodality in forest types generated from GFBi data. Colors reflect the value of the bimodality index (BI), with red representing BIs < −0.22 (deciduous-dominated forest clusters), blue representing BIs > 0.22 (evergreen-dominated forest clusters), yellow representing BIs from −0.22 – 0 (bistable-deciduous forest clusters) and cyan representing BIs from 0–0.22 (bistable-evergreen forest clusters). **B** Projected map of bimodality in forest types across the Northern Hemisphere based on random forest modelling. Colors reflect the projected value of the bimodality index (BI), and share the same scale as in **A**. Predictions in B were made for forest regions (1) above 15 degrees northern latitude, where > 98% of the GFBi data are located, (2) whose environmental conditions well represented by our training data (> 90% interpolation, see Fig. S9A). **C** Relative frequencies of plot-level relative evergreen abundance in deciduous-dominated, bistable-deciduous, bistable-evergreen and evergreen-dominated regions. Source data are provided as a Source Data file.

been suggested as a critical threshold for transitions in soil properties and ecosystem states[7,42]. Our analysis supports this idea, showing that forest phenological composition changes abruptly at soil pH values of 4.5–5.0 (Fig. S14).

While our findings provide valuable insights into alternative forest states, it is important to address several potential limitations:

1. Human influence: The possibility that human management[28], such as monoculture plantations, might induce bimodal patterns cannot be entirely dismissed. However, even after excluding plots with signs of recent human disturbances like harvesting or monoculture presence, the distribution of leaf types remained bimodally distributed (Figs. 1 & S3). Additionally, the positive demographic feedback observed in multi-year forest inventory data suggests that human interference alone does not account for these patterns (Figs. 2 & S6).

2. Categorization of leaf phenotypes: The oversimplification in classifying species strictly as evergreen or deciduous ignores the variety in leaf-shedding behaviors among species. For instance, trees that shed leaves for two months were classified similarly to those shedding for 10 months. However, our analysis confirmed

**A. Global analysis**

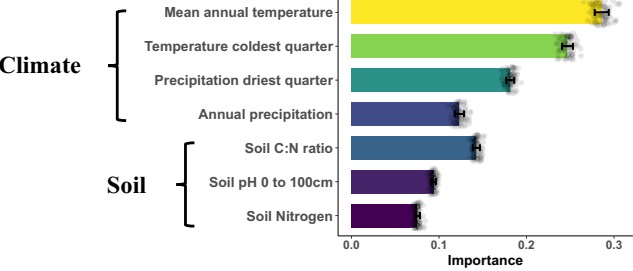

**B. Cluster-level analysis**

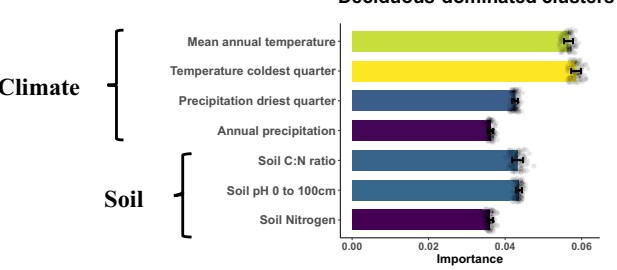
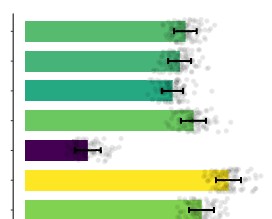
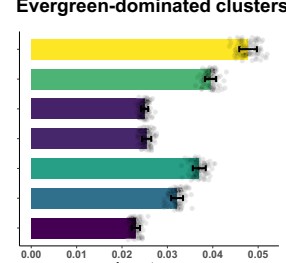

**Fig. 5 | Determinant analysis showing the variable importance based on random forest models.** Seven key covariates of forest leaf phenology composition were used to train the random forest models, whereby we ran each model 100 times on 100 bootstrapping training sets and then computed the mean and standard deviation of the variable permutation importance. **A**, Variable permutation importance (mean ± std, $n = 100$, individual data points are overlayed on the bar charts) for global random forest analysis using the bimodality index as response variable. Variables along y-axis in **A** are ordered by their mean importance. The continuous color scale represents the variable importance from high (yellow) to low (dark blue). **B**, Variable permutation importance (mean ± std, $n = 100$) for random forest analysis of plots within deciduous-dominated forests (left panel), bimodal forests (middle panel, including both bistable-deciduous and bistable evergreen forests) and evergreen-dominated forests (right panel). Analyses in panel **B** all use plot-level relative evergreen abundance as response variable. Source data are provided as a Source Data file.

that treating deciduousness as either a continuous or categorical trait does not alter its bimodal distribution among species (Fig. S33 & Fig. 1).

3. Geographic limitations: Our study is constrained by the risks of extrapolating findings to tropical regions and the Southern Hemisphere due to data scarcity. Therefore, we have confined our predictions and conclusions to forest regions primarily in the Northern Hemisphere above 15 degrees latitude, where over 98% of our dataset originates, primarily from North America and Europe. This limitation helps minimize risks but does not completely eliminate the possibility of geographic extrapolation errors in regions like Siberia and Asia, where tree species may have different evolutionary histories.

4. Data Interpolation: The soil layers included in our random forest models were interpolated from point observations and may thus correlate with climate variables. Although our model predictions are not affected by multicollinearity[43], this could influence the variable importance of climate and soil features. However, re-examining this using the World Soil Information Service (WOSIS) dataset[44], which includes local soil observations, confirmed that soil chemical variables more accurately explain variations in leaf phenology within bistable forest clusters than climate variables (Figs. S22–24).

5. Simulation limitations: Our 2000-year simulations were based on 5-year demographic trends from the FIA data, which may not capture long-term outcomes. However, the simulations, whether including or excluding feedbacks, start to diverge after only a few years (Figs. S17–19), providing evidence that short-term trends reflect longer-term patterns. Furthermore, our feedback model simulations did not include specific feedback mechanisms like plant-soil interactions but instead used plot-level relative evergreen abundance as a predictor for frequency-dependent feedback. This approach supports the notion that positive feedback can sustain bimodality, though more empirical research is needed to pinpoint specific feedback mechanisms.

In conclusion, our study reveals complementary lines of evidence to support the existence of alternative stable states and highlights regions in which bimodality in forest types is likely to occur. Given the close connection between forest types and ecosystem biogeochemical processes, our findings can improve our understanding of the occurrence of evergreen and deciduous forests, terrestrial carbon sequestration, and ecosystem feedbacks to the climate system.

## Methods

### Data preprocessing

**Forest inventory data.** Data for the North American continental analysis came from the US Forest Service's Forest Inventory and Analysis database v.9 (FIA)[32]. To eliminate the effects of management, we excluded plantations and actively managed or harvested forest plots from the analysis. To reduce the effects of small plots (plots with few trees are more likely to have only one leaf phenology type), we only included plots with at least ten individual trees. We also excluded plots where >50% of trees had died between census intervals[30] to exclude effects of major pest outbreaks or other disturbances. We assigned all observed trees as evergreen or deciduous based on species-level or genus-level leaf phenology data from the TRY database[34]. We only kept plots for which information on leaf phenology strategy was available for > 90% of trees (weighted by basal area). Our final FIA dataset included 45,276 unique forest sites, each with ~5-yr census intervals (the average time period for these observations is 2010 ~ 2015), along with information on environmental covariates.

For the analysis of European forests, we used inventory data from Germany, Spain, Sweden, Finland, and Belgium from the FunDivEUROPE[45] database and data from Switzerland from the Swiss Federal Institute for Forest, Snow and Landscape Research (WSL). As for the FIA analysis, leaf phenology type (evergreen versus deciduous) was assigned to each tree, and only plots for which leaf phenology information was available for >90% of the total basal area were kept. Due to the lack of management information for many plots, we accounted for the potential impact of human management on the establishment of monocultures by excluding plots 1) with fewer than ten tree individuals, 2) with only one species, 3) in which the relative basal area of a species was larger than 90% of the total cumulative basal area of all individuals in that plot. Our final European dataset included 15,431 unique forest sites, each with ~10-yr census intervals, along with information on environmental covariates.

Data for the global analysis came from the Global Forest Biodiversity Initiative (GFBi)[33]. Similarly, leaf phenology information was assigned to each tree, and only plots for which leaf phenology information was available for >90% of the total basal area were kept. The same filter as in the FunDivEUROPE data were applied to control for the effects of management and plot size. Most GFBi plots were only measured once, and for plots with timeseries data, the latest observation year was included in the analysis. The final GFBi dataset included 815,578 unique forest plots. The area of all plots is standardized to one hectare.

For all datasets, we computed two types of plot-level relative evergreen abundance (relEV) using stem density and basal area:

$$relEV_{area} = \frac{basalArea_{Evergreen}}{basalArea_{Evergreen} + basalArea_{Deciduous}} \quad (1)$$

$$relEV_{density} = \frac{stemDensity_{Evergreen}}{stemDensity_{Evergreen} + stemDensity_{Deciduous}} \quad (2)$$

The individual-based relative evergreen abundance ($relEV_{density}$) was used in the analysis presented in Fig. 1 and Fig. S3, which compares stem densities between the null model and the observations. All other analyses use area-based relative evergreen abundance ($relEV_{area}$). The two abundance types are highly similar ($R^2 = 0.93$, Fig. S7), and thus result in the same patterns.

**Environmental covariates.** We extracted spatial data on 62 commonly used environmental covariates, reflecting variations in climate, soil, topography, and human characteristics (see Table. S2). These covariates were used to train spatial random forest models for mapping. Of these variables, 53 were used to control for environmental filtering when testing for alternative stable states in leaf phenology strategies, excluding nine soil-related factors such as nitrogen density, C:N ratio, clay content, and soil pH, which are likely to drive positive feedbacks and generate alternative stable states. All covariate layers were standardized to EPSG:4326 (WGS84) projection at 30 arc-sec resolution (~1 km² at the equator). To reduce feature dimensionality, we conducted a principal component analysis on the 53 environmental covariates and selected the first ten principal components which captured 80% of the variation in the original 53 covariates. A summary of the leading ten principal components can be found in Fig. S10. These ten principal components were then used as predictors in all statistical models of tree growth, recruitment and survival for both the FIA and GFBi datasets.

**Bimodality testing**
If alternative stable states exist, the distribution of relative evergreen abundance should be bimodal. However, bimodal patterns can also result from environmental effects on the abundances of individual tree types or simply be due to overdispersion in the distribution of trees. If the bimodal patterns are purely driven by environmental conditions without overdispersion or biotic interactions such as con-phenological feedback, then a Poisson distribution should well depict the distribution of tree density, assuming that trees are independent from each other.

To control for the effect of environmental filtering, we fit generalized additive models for location, scale and shape (GAMLSS) to the plot-level absolute abundance of evergreen and deciduous trees, respectively, using a zero-adjusted Poisson distribution (Fig. 1) via the 'gamlss' and 'gamlss.dist' packages[46] within R v.4.1.2. All parameters of the distribution were modelled via smoothing functions of the ten environmental principal components.

From our fitted models, we randomly extracted evergreen and deciduous abundances in each plot based on the plot-specific environmental conditions, using the functions "rZAP" for a zero-adjusted Poisson distribution. With those randomly drawn abundances we calculated the relative evergreen abundance in each plot, using only plots with at least ten trees, and generated a relative evergreen abundance distribution across plots as done for the observed data. By repeating this 1000 times, we obtained 1000 simulations of the expected relative evergreen abundance distribution, that now include environmental information (Poisson) at the plot spatial scale. By binning the relative abundance distribution, we could calculate the 2.5% and 97.5% quantiles for the frequency of each bin according to our 1000 simulations. We chose a bin width of 0.05 and compared the observed frequency in each bin to the predicted frequency. If the observed frequencies in the outer bins (0–0.05 and 0.95–1) are higher than the predicted frequencies (above the respective 97.5% quantile from model simulations), this indicates that other factors apart from environmental parameters shape the observed bimodality.

If the abundances of evergreen and deciduous trees show positive feedback, evergreen and deciduous plot-level abundances should be more anticorrelated than predicted by environment only. We used Spearman's Rank correlation coefficient[47] and compared the correlation in our observed data to the distribution of correlation coefficients obtained from the 1000 random draws from the models. If the observed correlation is more negative than the correlations obtained from those random draws (Fig. S3), this suggests that environmental conditions alone are not sufficient to explain the relationship between evergreen and deciduous forests.

**Testing whether demographic positive feedback can reinforce evergreen or deciduous-dominated states**
Demographic processes of growth, recruitment, and mortality of evergreen and deciduous trees fundamentally determine changes in forest leaf phenology composition. We used GAMs to model each of these processes as a function of environmental factors, individual tree and stand characteristics (tree size, stand stem density and stand basal area) and the relative evergreen abundance in the forest (our feedback predictor), to quantify the influence of con-phenological frequency dependence. To explore the sign and magnitude of con-phenological neighborhood effects, we kept all predictors except relative evergreen abundance constant and used the model to make predictions along gradients of relative evergreen abundance.

We used the FIA data for demographic modelling, because it has repeated measurements for each plot. Methods used to develop the following demographic models and simulations were adapted from Averill et al.[30].

**Modelling tree recruitment.** We fit separate GAMs to model recruitment of evergreen and deciduous trees at the plot level across the most recent 5-yr time interval for each forest site using a Poisson distribution. We defined recruits of each leaf phenology type as all corresponding individuals in the current census, which were not present

in the previous census. We modelled recruitment of evergreen or deciduous trees as a function of plot basal area, stem density, basal area of con-phenological trees, the ten environmental principal components, and the relEV within a plot (i.e., the con-phenological feedback predictor). We included the basal area of con-phenological trees within the plot as a term in addition to the con-phenological predictor to account for the fact that recruitment generally increases with the abundance of a focal species or the same phenological strategies within a given site. We included spatial clusters (extracted from a global 'fishing net' at 20 km resolution) as a random effect to control for potential spatial autocorrelation. For all continuous predictors, we fit splines with a penalized thin-plate spline regression method with a basis dimension of 5 to avoid overfitting. To elucidate differences in evergreen and deciduous recruitment in respective forest types, we used the fitted recruitment GAMs to predict evergreen/deciduous recruitment in an evergreen-dominated forest (relEV = 0.9) and a deciduous-dominated forest (relEV = 0.1) (Fig. 2C). All environmental covariates and stand properties were held at mean values, with only the relEV varying.

**Modelling tree growth.** We fit separate GAMs to model evergreen and deciduous tree growth at the individual tree level across the most recent 5-yr time intervals for each forest site. Within each model, we modelled current tree diameter as a function of an individual tree's previous diameter, total plot basal area, plot stem density, the ten environmental principal components, and the relative evergreen abundance within a plot (i.e., the con-phenological feedback predictor). We included spatial clusters (extracted from a global 'fishing net' at 20 km resolution) as a random effect to control for potential spatial autocorrelation. Again, for all continuous predictors, we fit splines with a penalized thin-plate spline regression method with a basis dimension of 5 to avoid overfitting. We then used the fitted growth GAMs to predict evergreen/deciduous growth in an evergreen-dominated forest (relEV = 1) and a deciduous-dominated forest (relEV = 0) (Fig. 2B), holding all environmental covariates and stand properties at their mean values.

**Modelling tree mortality/survival.** We fit separate GAMs to model evergreen and deciduous tree mortality at the individual tree level across the most recent 5-yr time interval for each forest site. Within each model, we modelled tree mortality probability as a function of an individual tree's previous diameter, total plot basal area, plot stem density, the ten environmental principal components and the relative evergreen abundance within a plot (i.e., the con-phenological feedback predictor). We included spatial clusters (extracted from a global 'fishing net' at 20 km resolution) as a random effect to control for potential spatial autocorrelation and used penalized thin-plate spline with a basis dimension of 5 for all continuous predictors. Results are visualized as survival probability (1 – mortality probability). We then used the fitted mortality GAMs to predict evergreen/deciduous survival in an evergreen-dominated forest (relEV = 1) and a deciduous-dominated forest (relEV = 0) (Fig. 2A), holding all environmental covariates and stand properties at their mean values.

**Demographic simulation 1: testing whether demographic positive feedbacks can generate alternative stable states.** The positive con-phenological neighborhood effects might not be sufficient to generate and maintain alternative stable states under demographic stochasticity, environmental variance, and disturbances. To test whether the observed demographic positive feedbacks are strong enough to generate and maintain the magnitude of bimodal patterns, we ran a series of demographic simulations based on empirically fit GAMs for growth, recruitment and mortality mentioned above. We initialized 1000 forest plots with varying relEV uniformly drawn from and allowed them to grow for 2000 years. We initialized each plot with 20 trees, each of

which with a diameter at breast height of 12.7 cm (the smallest tree measured in the FIA surveys). We ran two sets of simulations, both of which used the same GAMs of growth, mortality, and recruitment described above, with the 'null' model excluding the con-phenological feedback as predictor (i.e., relEV), and the 'feedback' model including this potential feedback. The null model is required to test whether factors other than positive feedbacks, such as demographic lags, disturbance, or environmental heterogeneity, can generate bimodal patterns in leaf phenology strategies, whereas the feedback model allows testing whether con-phenological feedbacks are sufficient to generate and maintain alternative stable states.

Simulated plot area was identical to the respective FIA survey plot area. We included stand-replacing disturbances (e.g., fire, hurricanes, etc.) at a probability of 0.0036 per year. This probably was based on the overall North American stand-replacing disturbance probability (0.009 per year) minus the stand replacement that is due to management (0.0054 per year)[48]. When disturbance occurred in a stand, the model reset to assume the initial 20 trees per plot. The model randomly assigned leaf phenology status to regenerating trees based on the stands' initial relEV before disturbance. We incorporated environmental heterogeneity by randomly drawing plot-level environmental conditions from observed values across forest plots used to fit the GAM models.

Our exploration within the feedback model did not target any specific feedback mechanisms, such as plant-soil or plant-microclimate feedback. Rather, we accounted for potential positive frequency-dependent feedbacks by including the effects of plot-level relEV on tree demography and forest succession. Essentially, the positive frequency-dependent feedbacks indicate that a higher proportion of evergreen trees will foster more evergreen-dominated forests, and the same holds for deciduous trees.

In the realm of dynamic models, a standard method for simulating feedback requires the integration of the variable of interest into their dynamic equations, symbolized as $\frac{dx}{dt} \sim f(x)$. In parallel, models able to induce bifurcation points also need to be nonlinear (introducing this feedback in a nonlinear manner). This way, these equations confront the two lines of complexity required for understanding feedbacks as generators of bifurcations (nonlinear relationships involving the variable of interest both as cause and consequence)[49].

We simulated this feedback through a set of equations relating growth, recruitment, and mortality to evergreen abundance. As mentioned above, we utilized generalized additive models to fit tree growth, recruitment, and survival as functions of the environment, stand properties, and relEV. Consequently, the functions can be represented as: $growth/recruitment/survival \sim f(relEV, \ldots)$. In each time step of the feedback simulation, evergreen abundance in a plot may change due to recruitment, growth and mortality of trees within that plot, hence: $\frac{d(relEV)}{dt} \sim f_1(growth, recruitment, survival) \sim f_2(relEV)$.

This approach accomplishes adding a feedback, as it maintains the form $\frac{d(relEV)}{dt} \sim f(relEV)$. Moreover, all equations are grounded in generalized additive models, which are nonlinear.

**Demographic simulation 2: testing whether demographic positive feedbacks can maintain alternative stable states.** To test whether feedbacks can maintain alternative stable states, we ran the second group of simulations with the same setting as for the demographic simulation 1, but with a bimodal initialization. Specifically, we initialized the 1000 forest plots as 500 purely evergreen forests and 500 purely deciduous plots to mimic the extreme bimodal scenario.

**Demographic simulation 3: testing for hysteresis during phenological transition across soil pH gradients.** The existence of hysteresis during ecosystem transition along environmental gradients is one of the key signals for the presence of alternative stable states. To test for hysteresis, we first fit GAM demographic models (recruitment, growth

and mortality) for both evergreen and deciduous trees using the method described above, but added mean annual temperature (MAT) as an extra predictor, as it is a major driver of the transition between evergreen and deciduous forests[7,50]. Based on these GAMs, we ran two sets of demographic simulations (as in the demographic simulation 1) across a gradient of MAT from −2 °C to 23 °C, which is the empirical range of MAT in forest plots included in the US FIA dataset. In the first set of simulations, we assumed that 80% of forest plots (800 of 1000) are initially covered 100% with evergreen trees, while the remaining 20% (200 plots) were simulated to consist solely of deciduous trees. In the second set of simulations, 800 forest plots were initialized as purely deciduous plots and the remainder as purely evergreen plots. We ran each set of simulations under both 'null' model (excluding feedback predictors) and 'feedback' scenarios, and for 2000 years to minimize the influence of demographic lags on model outcomes.

### Random forest modelling to map bimodality in leaf phenology strategies

To quantify the extent of bimodality in the distribution of leaf phenology strategies, as well as the spatial variation therein, we developed two independent random forest models with different plot-partitioning methods for global projection (see graphic demonstration in Fig. S8). In the first partitioning method ("spatial clustering" approach), we partitioned the global forest zones using a 'fishing net' with 10 arc-min (~20 km) grid size. We removed clusters with fewer than 10 forest plots, resulting in 14,931 clusters. For each cluster, we aggregated forest composition information of all 1-hectare GFBi forest plots to calculate a bimodality index (BI), which allowed us to quantify bimodality in the leaf phenology distribution across plots. In the second partitioning method ("environmental clustering" approach), we implemented K-means clustering[51] to group forest plots into 15,000 clusters (close to the sample size of spatial clustering) based on the leading 3 environmental principal components of each plot. We removed clusters with fewer than 10 forest plots, resulting in 14,858 clusters. Similarly, for each cluster, we aggregated plot-level relative evergreen abundance to calculate the BI.

The spatial clustering approach represents patterns resulting from spatial processes, but cannot account for residual environmental variation within clusters. By contrast, the environmental clustering approach ensures that each cluster consists of environmentally homogenous plots, while plots within clusters are not necessarily in spatial proximity. Correlation between the predictions from the spatial random forests and the predictions from the environmental random forest models revealed high agreement between the two approaches, with an $R^2$ of 0.72. The use of both methods allows us to test for the sensitivity of the inferred patterns to these methodological considerations, ultimately increasing our confidence in the global predictions.

To calculate the BI, we first calculated Hartigan's dip test[52] statistic $D$ for the distribution of relative evergreen abundance in a grid using the 'diptest' R package, which represents the maximum distance of an empirical distribution to the best fitting unimodal distribution. We then computed the adapted dip statistic[52] as:

$$D' = \sqrt{n}D \tag{3}$$

where n represents the sample size and $D$ the dip test statistic. This metric is negatively correlated with the $P$ values of the dip-test, because the larger the statistical distance of a distribution from unimodality, the less likely the distribution is to be unimodal. When $n$ tends to large values, $D' > 0.546$ indicates significant multimodality ($p < 0.05$). For our data, in which grid cells ary in $n$, we chose $D' > 0.5$ as the threshold because all grids with $D' > 0.5$ have $p < 0.05$. $D'$ can only differentiate bimodal from unimodal distributions, but cannot indicate which side a unimodal distribution is skewed toward, which is

important in our case as we were interested in whether evergreen or deciduous trees dominate in case of a unimodal distribution. Therefore, we also calculated the skewness $S$ for the distribution of relative evergreen abundance. In a unimodal distribution, $S < 0$ represents evergreen dominance, whereas $S > 0$ represents deciduous dominance. The bimodality index BI was then defined as:

$$BI = -e^{-aD'^b} \times sign(S) \tag{4}$$

where $a$ and $b$ are parameters to control the range of BI. We set $a$ to 6 and $b$ to 2, so that the BI across all grids ranged from −1 to 1, whereby BIs < −0.22 represent deciduous-dominated grids, BIs from −0.22 to 0 represent bistable deciduous grids, BIs from 0 to 0.22 represent bistable evergreen grids, and BIs > 0.22 represent evergreen-dominated grids. The ±0.22 cutoff was directly computed from the cutoff of $D'$ (0.5) using Eq. (4). Different combinations of a and b will give different ranges of BI as well as different threshold values, but will not change the final patterns. To predict spatial variation in BI, we then trained random forest models using 62 environmental predictors, covering climate, soil, topography, vegetation and human activity characteristics (Table. S2). Finally, we predicted the BI (using EPSG 8857 equal-earth projection) across global forest regions in which environmental conditions were well represented by our training data (>90% interpolation).

### The relative importance of climate versus soil feedbacks

The observed positive feedbacks that generate and maintain alternative stable states likely affect leaf phenology strategies at the local scale where trees facilitate the establishment of their own phenology type by engineering soil conditions. By contrast, the large-scale distribution of leaf phenology is largely constrained by climate and the physiological limits of evergreen and deciduous trees[7]. We, therefore, hypothesized that (1) at large spatial scales (e.g., global), climatic factors are the main drivers of the presence or absence of alternative stable states in ecosystems, (2) at local scales, i.e., within grid cells with bimodal leaf phenology distributions, soil properties that might drive the positive feedbacks co-explain variation in the relative abundance of evergreen forests, and (3) in evergreen- or deciduous-dominated grids without bimodal leaf phenology distributions, climatic factors again are more important than edaphic factors.

To test these hypotheses, we trained two sets of random forest models using the spatially partitioned GFBi data. The first random forest model predicted forest bimodality as a function of seven key determinants, namely mean annual temperature, annual precipitation, mean temperature of the coldest quarter, precipitation of the driest quarter, soil pH 0 to 100 cm, soil nitrogen density and soil C:N, at the global scale. The second set of models predicted relative evergreen abundance within plots as a function of the seven determinants for either all plots within bimodal (BI = −0.22–0.22), evergreen-dominated (BI > 0.22) or deciduous-dominated (BI < −0.22) forest grids (from the "spatial clustering" approach). For each model, we then calculated the permutation importance of each variable. We trained these models using 100 bootstrap samples, with a sampling proportion of 33% relative to the training data, and then checked the permutation importance (mean ± SD) of each variable used in the random forest models (Fig. 5).

### Reporting summary

Further information on research design is available in the Nature Portfolio Reporting Summary linked to this article.

## Data availability

The GFBi data is available upon request via Science-i (https://science-i.org/) or the GFBI website (https://www.gfbinitiative.org/). The FIA data are publicly available from the FIA datamart (https://apps.fs.usda.

gov/fia/datamart/). The FunDivEUROPE data is available upon request via http://project.fundiveurope.eu. All environmental covariates are publicly available and detailed in Table S2. Source data to reproduce the figures are provided as a Source Data file[53] (https://zenodo.org/records/11048857).

## Code availability

The code used for this study is available at https://github.com/Yibiaozou/AltSS_ForestLeafPhenology and Zenodo[54] (https://doi.org/10.5281/zenodo.11035706).

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

## Acknowledgements

We warmly thank all members of the Crowther lab team, including those not listed as coauthors, for their invaluable support. The collaboration and development of the manuscript were supported by the web-based science platform, science-i.org. We thank the Global Forest Biodiversity Initiative (GFBI) for establishing the data standards and collaborative framework. This work was supported by grants to TWC from the Bernina Foundation and DOB Ecology, and CMZ from the Ambizione Fellowship program (#PZ00P3_193646). MB was supported by a Ramón y Cajal grant (RYC2021-031797-I) from the Spanish Ministry of Sciences. The European forest inventory data was provided by the FunDivEUROPE project, which received funding from the European Union Seventh Framework Programme (FP7/2007-2013) under grant agreement no 265171. We thank the Swiss Federal Institute for Forest, Snow and Landscape Research (WSL), Birmensdorf, and Dr. Christian Temperli for providing Swiss National Forest Inventory (LFI) data for the periods 1983-85, 1993-95, 2004-06, and 2009-2017. This study was also supported by the TRY initiative that is maintained by the Max Planck Institute for Biogeochemistry in Jena, Germany, and currently supported by DIVERSITAS/Future Earth and the German Centre for Integrative Biodiversity Research (iDiv) Halle-Jena-Leipzig. National Natural Science Foundation of China (31800374). The ReVaTene dataset is funded by the Education and Research Ministry of Côte d'Ivoire, as part of the Debt Reduction-Development Contracts (C2Ds) managed by IRD. JCS considers this work a contribution to his VILLUM Investigator project "Biodiversity Dynamics in a Changing World", funded by VILLUM FONDEN (grant 16549), and Center for Ecological Dynamics in a Novel Biosphere (ECONOVO), funded by the Danish National Research Foundation (grant DNRF173). AFS considers this work a contribution to his UFRN project PVA12722-2015, and is thankful to Solon J. Longhi for data sharing (Conselho Nacional de Desenvolvimento Científico e Tecnológico 520053/1998-2). ALG considers this work a contribution to his IFFSC project (FAPESC, SDE, IMA, and CNPq grants). VS thanks for support to APVV 20-0168 from the Slovak Research and Development Agency. We are thankful to the State of São Paulo Research Foundation (FAPESP) for supporting the Atlantic Forest plots through the BIOTA/FAPESP Program (Project Functional Gradient 2003/12595-7, 2010/20811-7 & ECOFOR 2012/51872-5), and the Brazilian National Research Council (CNPq grant 403710/2012-0). FCT – Portuguese Foundation for Science and Technology, project UIDB/04033/2020 and ICNF-Instituto da Conservação da Natureza. RAINFOR plots here were supported by a major grant from the Gordon and Betty Moore Foundation. We also thank NERC for long-term support of RAINFOR and ForestPlots.net (including NE/X014347/1, NE/N012542/1, NE/N012542/1, NE/X014347/1) and additional sources to O.L.P. including the ERC (Advanced Grant 291585, T-FORCES). P.R.B. and M.A.Z. acknowledge funding from the CLIMB-FOREST Horizon Europe Project (No 101059888) that was funded by the European Union.

## Author contributions

Y.Z. conceived, developed, and wrote the paper with assistance from C.M.Z. and T.W.C., Y.Z. performed the analyses with assistance from C.A., C.M.Z., H.M., J.M., M.B. and L.M. T.W.C. and L.B.M. helped with data coordination. The members of the GFBi consortium provided the global forest inventory data. All authors reviewed and provided input on the manuscript.

## Competing interests

The authors declare no competing interests.

## Additional information

[1]Institute of Integrative Biology, ETH Zurich (Swiss Federal Institute of Technology), Universitätsstrasse 16, 8092 Zurich, Switzerland. [2]Department of Global Ecology, Carnegie Institution for Science, Stanford, CA, USA. [3]Swiss Federal Institute for Forest, Snow and Landscape Research WSL, 8903 Birmensdorf, Switzerland. [4]Department of Forestry and Natural Resources, Purdue University, West Lafayette, IN, USA. [5]Department of Agricultural and Forest Sciences and Engineering, University of Lleida, Lleida, Spain. [6]Forest Science and Technology Centre of Catalonia (CTFC), Solsona, Spain. [7]Wageningen University and Research, Wageningen, The Netherlands. [8]Department of Forest Resources, University of Minnesota, St. Paul, MN, USA. [9]Hawkesbury Institute for the Environment, Western Sydney University, Penrith, NSW, Australia. [10]Institute for Global Change Biology, and School for Environment and Sustainability, University of Michigan, Ann Arbor, MI, USA. [11]Plant Physiology work group, Estonian University of Life Sciences, Tartu, Estonia. [12]Department of Forest Resource Management, Swedish University of Agricultural Sciences, Umeå, Sweden. [13]Forstliche Versuchs- und Forschungsanstalt Baden-Württemberg, Freiburg im Breisgau, Germany. [14]NBN Trust, Nottingham, UK. [15]Universidad de Alcalá, Alcala de Henares, Spain. ✉e-mail: yibiao.zou@usys.ethz.ch

## GFBI consortium

Meinrad Abegg[3], Yves C. Adou Yao[16], Giorgio Alberti[17,18], Angelica M. Almeyda Zambrano[19], Braulio Vilchez Alvarado[20], Esteban Alvarez-Dávila[21], Patricia Alvarez-Loayza[22], Luciana F. Alves[23], Christian Ammer[24], Clara Antón-Fernández[25], Alejandro Araujo-Murakami[26], Luzmila Arroyo[26], Valerio Avitabile[27], Gerardo A. Aymard[28,29], Timothy R. Baker[30], Radomir Bałazy[31], Olaf Banki[32], Jorcely G. Barroso[33], Meredith L. Bastian[34,35], Jean-Francois Bastin[36], Luca Birigazzi[37], Philippe Birnbaum[38,39,40], Robert Bitariho[41], Pascal Boeckx[42], Frans Bongers[7], Olivier Bouriaud[43], Pedro H. S. Brancalion[44], Susanne Brandl[45], Francis Q. Brearley[46], Roel Brienen[30], Eben N. Broadbent[47], Helge Bruelheide[48,49], Filippo Bussotti[50], Roberto Cazzolla Gatti[51], Ricardo G. César[44], Goran Cesljar[52], Robin Chazdon[53,54], Han Y. H. Chen[55], Chelsea Chisholm[1], Hyunkook Cho[56], Emil Cienciala[57,58], Connie Clark[59], David Clark[60], Gabriel D. Colletta[61], David A. Coomes[62], Fernando Cornejo Valverde[63], José J. Corral-Rivas[64], Philip M. Crim[65,66], Jonathan R. Cumming[65], Selvadurai Dayanandan[67], André L. de Gasper[68], Mathieu Decuyper[69], Géraldine Derroire[70], Ben DeVries[71], Ilija Djordjevic[72], Jiri Dolezal[73,74], Aurélie Dourdain[70], Nestor Laurier Engone Obiang[75], Brian J. Enquist[76,77], Teresa J. Eyre[78], Adandé Belarmain Fandohan[79], Tom M. Fayle[80,81], Ted R. Feldpausch[82], Leandro V. Ferreira[83], Leena Finér[84], Markus Fischer[85], Christine Fletcher[86], Jonas Fridman[87], Lorenzo Frizzera[88], Javier G. P. Gamarra[89], Damiano Gianelle[88], Henry B. Glick[90], David J. Harris[91], Andrew Hector[92], Andreas Hemp[93], Geerten Hengeveld[7], Bruno Hérault[94,95], John L. Herbohn[54], Martin Herold[96], Annika Hillers[97,98], Eurídice N. Honorio Coronado[99], Cang Hui[100,101], Thomas Ibanez[102], Amaral Iêda[103], Nobuo Imai[104], Andrzej M. Jagodziński[105,106], Bogdan Jaroszewicz[107], Vivian Kvist Johannsen[108], Carlos A. Joly[109], Tommaso Jucker[110], Ilbin Jung[56], Viktor Karminov[111], Kuswata Kartawinata[112], Elizabeth Kearsley[113], David Kenfack[114], Deborah K. Kennard[115], Sebastian Kepfer-Rojas[108], Gunnar Keppel[116], Mohammed Latif Khan[117], Timothy J. Killeen[26], Hyun Seok Kim[118,119,120,121], Kanehiro Kitayama[122], Michael Köhl[123], Henn Korjus[124], Florian Kraxner[125], Diana Laarmann[124], Mait Lang[124], Simon L. Lewis[30,126], Huicui Lu[127], Natalia V. Lukina[128], Brian S. Maitner[76], Yadvinder Malhi[129], Eric Marcon[130], Beatriz Schwantes Marimon[131], Ben Hur Marimon-Junior[131], Andrew R. Marshall[54,132,133], Emanuel H. Martin[134], Dmitry Kucher[135], Jorge A. Meave[136], Omar Melo-Cruz[137], Casimiro Mendoza[138], Cory Merow[53], Abel Monteagudo Mendoza[139,140], Vanessa S. Moreno[44], Sharif A. Mukul[141,142], Philip Mundhenk[123], María Guadalupe Nava-Miranda[143,144], David Neill[145], Victor J. Neldner[78], Radovan V. Nevenic[72], Michael R. Ngugi[78], Pascal A. Niklaus[146], Jacek Oleksyn[105], Petr Ontikov[111], Edgar Ortiz-Malavasi[20], Yude Pan[147], Alain Paquette[148], Alexander Parada-Gutierrez[26], Elena I. Parfenova[149], Minjee Park[4,118], Marc Parren[150], Narayanaswamy Parthasarathy[151], Pablo L. Peri[152], Sebastian Pfautsch[153], Oliver L. Phillips[30], Nicolas Picard[154], Maria Teresa T. F. Piedade[155], Daniel Piotto[156], Nigel C. A. Pitman[112], Irina Polo[157], Lourens Poorter[7], Axel D. Poulsen[91], John R. Poulsen[59,158], Hans Pretzsch[159], Freddy Ramirez Arevalo[160], Zorayda Restrepo-Correa[161], Mirco Rodeghiero[88,162], Samir G. Rolim[156], Anand Roopsind[163], Francesco Rovero[164,165], Ervan Rutishauser[166], Purabi Saikia[167], Christian Salas-Eljatib[168,169,170], Philippe Saner[171], Peter Schall[24], Mart-Jan Schelhaas[7], Dmitry Schepaschenko[125,172], Michael Scherer-Lorenzen[173], Bernhard Schmid[146], Jochen Schöngart[155], Eric B. Searle[148], Vladimír Seben[174], Josep M. Serra-Diaz[53,175], Douglas Sheil[150,176], Anatoly Z. Shvidenko[125,128], Javier E. Silva-Espejo[177], Marcos Silveira[178], James Singh[179], Plinio Sist[94], Ferry Slik[180], Bonaventure Sonké[181], Alexandre F. Souza[182], Stanislaw Miscicki[183], Krzysztof J. Stereńczak[31], Jens-Christian Svenning[184], Miroslav Svoboda[185], Ben Swanepoel[186], Natalia Targhetta[155], Nadja Tchebakova[149], Hans ter Steege[32,187], Raquel Thomas[188], Elena Tikhonova[128], Peter M. Umunay[189], Vladimir A. Usoltsev[190], Renato Valencia[191], Fernando Valladares[192], Fons van der Plas[193], Tran Van Do[194], Michael E. van Nuland[195], Rodolfo M. Vasquez[139], Hans Verbeeck[196], Helder Viana[197,198], Alexander C. Vibrans[68,199], Simone Vieira[200], Klaus von Gadow[201], Hua-Feng Wang[202], James V. Watson[203], Gijsbert D. A. Werner[204], Bertil Westerlund[87], Susan K. Wiser[205], Florian Wittmann[206], Hannsjoerg Woell[207], Verginia Wortel[208], Roderik Zagt[209], Tomasz Zawiła-Niedźwiecki[210], Chunyu Zhang[211], Xiuhai Zhao[211], Mo Zhou[4], Zhi-Xin Zhu[202] & Irie C. Zo-Bi[95]

[16]UFR Biosciences, University Félix Houphouët-Boigny, Abidjan, Côte d'Ivoire. [17]Department of Agricultural, Food, Environmental and Animal Sciences, University of Udine, Udine, Italy. [18]Faculty of Science and Technology, Free University of Bolzano, Bolzano, Italy. [19]Spatial Ecology and Conservation

Laboratory, Department of Tourism, Recreation and Sport Management, University of Florida, Gainesville, FL 32611, USA. [20]Forestry School, Tecnológico de Costa Rica TEC, Cartago, Costa Rica. [21]Fundacion ConVida, Universidad Nacional Abierta y a Distancia, UNAD, Medellin, Colombia. [22]Field Museum of Natural History, Chicago, USA. [23]Center for Tropical Research, Institute of the Environment and Sustainability, UCLA, Los Angeles, CA, USA. [24]Silviculture and Forest Ecology of the Temperate Zones, University of Göttingen, Göttingen, Germany. [25]Division of Forest and Forest Resources, Norwegian Institute of Bioeconomy Research (NIBIO), Ås, Norway. [26]Museo de Historia natural Noel kempff Mercado, Santa Cruz, Bolivia. [27]European Commission, Joint Research Center, Ispra, Italy. [28]UNELLEZ-Guanare, Programa de Ciencias del Agro y el Mar, Herbario Universitario (PORT), Portuguesa, Venezuela. [29]Compensation International S. A. Ci Progress-GreenLife, Bogotá, DC, Colombia. [30]School of Geography, University of Leeds, Leeds, UK. [31]Department of Geomatics, Forest Research Institute, Raszyn, Poland. [32]Naturalis Biodiversity Center, Leiden, The Netherlands. [33]Centro Multidisciplinar, Universidade Federal do Acre, Rio Branco, Brazil. [34]Proceedings of the National Academy of Sciences, Washington, DC, USA. [35]Department of Evolutionary Anthropology, Duke University, Durham, NC 27708, USA. [36]TERRA Teach and Research Centre, Gembloux Agro Bio-Tech, University of Liege, Liege, Belgium. [37]Forestry Consultant, Via Unione Sovietica 105, 58100 Grosseto, Italy. [38]Institut Agronomique néo-Calédonien (IAC), Nouméa, New Caledonia. [39]AMAP, Univ Montpellier, Montpellier, France. [40]CIRAD, CNRS, INRAE, IRD, Montpellier, France. [41]Institute of Tropical Forest Conservation, Mbarara University of Sciences and Technology, Mbarara, Uganda. [42]Isotope Bioscience Laboratory - ISOFYS, Ghent University, Ghent, Belgium. [43]Integrated Center for Research, Development and Innovation in Advanced Materials, Nanotechnologies, and Distributed Systems for Fabrication and Control (MANSiD), Stefan cel Mare University of Suceava, Suceava, Romania. [44]Department of Forest Sciences, Luiz de Queiroz College of Agriculture, University of São Paulo, Piracicaba, Brazil. [45]Bavarian State Institute of Forestry, Freising, Germany. [46]Department of Natural Sciences, Manchester Metropolitan University, Manchester, UK. [47]Spatial Ecology and Conservation Laboratory, School of Forest Resources and Conservation, University of Florida, Gainesville, FL 32611, USA. [48]Institute of Biology, Geobotany and Botanical Garden, Martin Luther University Halle-Wittenberg, Halle-, Wittenberg, Germany. [49]German Centre for Integrative Biodiversity Research (iDiv) Halle-Jena-Leipzig, Leipzig, Germany. [50]Department of Agriculture, Food, Environment and Forest (DAGRI), University of Firenze, Florence, Italy. [51]Department of Biological, Geological, and Environmental Sciences, University of Bologna, Bologna, Italy. [52]Department of Spatial Regulation, GIS and Forest Policy, Institute of Forestry, Belgrade, Serbia. [53]Department of Ecology and Evolutionary Biology, University of Connecticut, Storrs, CT, USA. [54]Forest Research Institute, University of the Sunshine Coast, Sippy Downs, Queensland, QLD, Australia. [55]Faculty of Natural Resources Management, Lakehead University, Thunder Bay, ON, Canada. [56]Division of Forest Resources Information, Korea Forest Promotion Institute, Seoul, South Korea. [57]IFER - Institute of Forest Ecosystem Research, Jilove u Prahy, Czech Republic. [58]Global Change Research Institute CAS, Brno, Czech Republic. [59]Nicholas School of the Environment, Duke University, Durham, NC, USA. [60]Department of Biology, University of Missouri-St Louis, St. Louis, MO, USA. [61]Programa de Pós-graduação em Biologia Vegetal, Instituto de Biologia, Universidade Estadual de Campinas, Campinas, Brazil. [62]Department of Plant Sciences and Conservation Research Institute, University of Cambridge, Cambridge, UK. [63]Andes to Amazon Biodiversity Program, Madre de Dios, Peru. [64]Facultad de Ciencias Forestales y Ambientales, Universidad Juárez del Estado de Durango, Durango, Mexico. [65]Department of Biology, West Virginia University, Morgantown, WV, USA. [66]Department of Physical and Biological Sciences, The College of Saint Rose, Albany, NY, USA. [67]Biology Department, Centre for Structural and Functional Genomics, Concordia University, Montreal, QC, Canada. [68]Natural Science Department, Universidade Regional de Blumenau, Blumenau, Brazil. [69]World Agroforestry (ICRAF), P.O. Box 30677, 00100 Nairobi, Kenya. [70]Cirad, UMR EcoFoG (AgroParisTech, CNRS, INRAE, Université des Antilles, Université de la Guyane), Campus Agronomique, Kourou, French Guiana. [71]Department of Geographical Sciences, University of Maryland, College Park, MD, USA. [72]Institute of Forestry, Belgrade, Serbia. [73]Institute of Botany, The Czech Academy of Sciences, Třeboň, Czech Republic. [74]Department of Botany, Faculty of Science, University of South Bohemia, České Budějovice, Czech Republic. [75]IRET, Herbier National du Gabon (CENAREST), Libreville, Gabon. [76]Department of Ecology and Evolutionary Biology, University of Arizona, Tucson, AZ, USA. [77]The Santa Fe Institute, Santa Fe, NM, USA. [78]Queensland Herbarium, Department of Environment and Science, Toowong, QLD, Australia. [79]Ecole de Foresterie et Ingénierie du Bois, Université Nationale d'Agriculture, Kétou, Benin. [80]School of Biological and Behavioural Sciences, Queen Mary University of London, London, UK. [81]Biology Centre of the Czech Academy of Sciences, Institute of Entomology, Ceske Budejovice, Czech Republic. [82]Geography, College of Life and Environmental Sciences, University of Exeter, Exeter, UK. [83]Museu Paraense Emílio Goeldi. Coordenação de Ciências da Terra e Ecologia, Belém, Pará, Brasil. [84]Natural Resources Institute Finland (Luke), Joensuu, Finland. [85]Institute of Plant Sciences, University of Bern, Bern, Switzerland. [86]Forest Research Institute Malaysia, Kuala Lumpur, Malaysia. [87]Department of Forest Resource Management, Swedish University of Agricultural Sciences SLU, Umea, Sweden. [88]Research and Innovation Center, Fondazione Edmund Mach, San Michele All'adige, Italy. [89]Forestry Division, Food and Agriculture Organization of the United Nations, Rome, Italy. [90]Glick Designs LLC, Hadley, MA, USA. [91]Royal Botanic Garden Edinburgh, Edinburgh, UK. [92]Department of Plant Sciences, University of Oxford, Oxford, UK. [93]Department of Plant Systematics, University of Bayreuth, Bayreuth, Germany. [94]Cirad, UPR Forêts et Sociétés, University of Montpellier, Montpellier, France. [95]Department of Forestry and Environment, National Polytechnic Institute (INP-HB), Yamoussoukro, Côte d'Ivoire. [96]Helmholtz GeoResearch Center (GFZ), Potsdam, Germany. [97]Centre for Conservation Science, The Royal Society for the Protection of Birds, Sandy, UK. [98]Wild Chimpanzee Foundation, Liberia Office, Monrovia, Liberia. [99]Instituto de Investigaciones de la Amazonía Peruana, Iquitos, Peru. [100]Centre for Invasion Biology, Department of Mathematical Sciences, Stellenbosch University, Stellenbosch, South Africa. [101]Theoretical Ecology Unit, African Institute for Mathematical Sciences, Cape Town, South Africa. [102]AMAP, Univ Montpellier, CIRAD, CNRS, INRAE, IRD, Montpellier, France, Montpellier, France. [103]National Institute of Amazonian Research, Manaus, Brazil. [104]Department of Forest Science, Tokyo University of Agriculture, Tokyo, Japan. [105]Institute of Dendrology, Polish Academy of Sciences, Kórnik, Poland. [106]Poznań University of Life Sciences, Department of Game Management and Forest Protection, Poznań, Poland. [107]Faculty of Biology, Białowieża Geobotanical Station, University of Warsaw, Białowieża, Poland. [108]Department of Geosciences and Natural Resource Management, University of Copenhagen, Copenhagen, Denmark. [109]Department of Plant Biology, Institute of Biology, University of Campinas, UNICAMP, Campinas, Brazil. [110]School of Biological Sciences, University of Bristol, Bristol, UK. [111]Forestry Faculty, Bauman Moscow State Technical University, Moscow, Russia. [112]Field Museum of Natural History, Chicago, IL, USA. [113]CAVElab-Computational and Applied Vegetation Ecology, Department of Environment, Ghent University, Ghent, Belgium. [114]CTFS-ForestGEO, Smithsonian Tropical Research Institute, Balboa, Panama. [115]Department of Physical and Environmental Sciences, Colorado Mesa University, Grand Junction, CO, USA. [116]UniSA STEM and Future Industries Institute, University of South Australia, Adelaide, SA, Australia. [117]Department of Botany, Dr Harisingh Gour Vishwavidyalaya (A Central University), Sagar 470003, India. [118]Department of Agriculture, Forestry and Bioresources, Seoul National University, Seoul, South Korea. [119]Interdisciplinary Program in Agricultural and Forest Meteorology, Seoul National University, Seoul, South Korea. [120]National Center for Agro Meteorology, Seoul, South Korea. [121]Research Institute for Agriculture and Life Sciences, Seoul National University, Seoul, South Korea. [122]Graduate School of Agriculture, Kyoto University, Kyoto, Japan. [123]Institute for World Forestry, University of Hamburg, Hamburg, Germany. [124]Institute of Forestry and Engineering, Estonian University of Life Sciences, Tartu, Estonia. [125]Ecosystems Services and Management, International Institute for Applied Systems Analysis, Laxenburg, Austria. [126]Department of Geography, University College London, London, UK. [127]Faculty of Forestry, Qingdao Agricultural University, Qingdao, China. [128]Center for Forest Ecology and Productivity, Russian Academy of Sciences, Moscow, Russia. [129]School of Geography, University of Oxford, Oxford, UK. [130]AgroParisTech, UMR-AMAP, Cirad, CNRS, INRA, IRD, Université de Montpellier, Montpellier, France. [131]Departamento de Ciências Biológicas, Universidade do Estado de Mato Grosso, Nova

Xavantina, Brazil. [132]Department of Environment and Geography, University of York, York, UK. [133]Flamingo Land Ltd, Kirby Misperton, UK. [134]Department of Wildlife Management, College of African Wildlife Management, Mweka, Tanzania. [135]Peoples Friendship University of Russia (RUDN University), Moscow, Russia. [136]Departamento de Ecología y Recursos Naturales, Facultad de Ciencias, Universidad Nacional Autónoma de México, Mexico City, Mexico. [137]Universidad del Tolima, Ibagué, Colombia. [138]Universidad Mayor de San Simón, Escuela de Ciencias Forestales, Colegio de Profesionales Forestales de Cochabamba, Cochabamba, Bolivia. [139]Jardín Botánico de Missouri, Pasco, Peru. [140]Universidad Nacional de San Antonio Abad del Cusco, Cusco, Peru. [141]Department of Environment and Development Studies, United International University, Dhaka 1212 Bangladesh, Sippy Downs, Queensland Bangladesh. [142]Centre for Research on Land-use Sustainability, Dhaka, Bangladesh. [143]Colegio de Ciencias y Humanidades. Universidad Juárez del Estado de Durango, Durango, Mexico. [144]Programa de doctorado en Ingeniería para el desarrollo rural y civil. Escuela de Doctorado Internacional de la Universidad de Santiago de Compostela (EDIUS), Santiago de Compostela, Spain. [145]Universidad Estatal Amazónica, Puyo, Pastaza, Ecuador. [146]Department of Geography, Remote Sensing Laboratories, University of Zürich, Zurich, Switzerland. [147]Climate, Fire, and Carbon Cycle Sciences, USDA Forest Service, Durham, NC, USA. [148]Centre for Forest Research, Université du Québec à Montréal, Montréal, QC, Canada. [149]V. N. Sukachev Institute of Forest, FRC KSC, Siberian Branch of the Russian Academy of Sciences, Krasnoyarsk, Russia. [150]Forest Ecology and Forest Management Group, Wageningen University & Research, Wageningen, The Netherlands. [151]Department of Ecology and Environmental Sciences, Pondicherry University, Puducherry, India. [152]Instituto Nacional de Tecnología Agropecuaria (INTA), Universidad Nacional de la Patagonia Austral (UNPA), Consejo Nacional de Investigaciones Científicas y Tecnicas (CONICET), Río Gallegos, Argentina. [153]School of Social Sciences (Urban Studies), Western Sydney University, Penrith, NSW, Australia. [154]Forestry Department, Food and Agriculture Organization of the United Nations, Rome, Italy. [155]Instituto Nacional de Pesquisas da Amazônia, Manaus, Brazil. [156]Laboratório de Dendrologia e Silvicultura Tropical, Centro de Formação em Ciências Agroflorestais, Universidade Federal do Sul da Bahia, Itabuna, Brazil. [157]Jardín Botánico de Medellín, Medellin, Colombia. [158]The Nature Conservancy, 2424 Spruce St., Boulder, CO 80302, USA. [159]Chair for Forest Growth and Yield Science, TUM School for Life Sciences, Technical University of Munich, Munich, Germany. [160]Universidad Nacional de la Amazonía Peruana, Iquitos, Peru. [161]Servicios Ecosistémicos y Cambio Climático (SECC), Fundación Con Vida & Corporación COL-TREE, Medellín, Colombia. [162]Centro Agricoltura, Alimenti, Ambiente, University of Trento, San Michele All'adige, Italy. [163]Department of Biological Sciences, Boise State University, Boise, ID, USA. [164]Department of Biology, University of Florence, Florence, Italy. [165]Tropical Biodiversity, MUSE - Museo delle Scienze, Trento, Italy. [166]Info Flora, Geneva, Switzerland. [167]Department of Environmental Sciences, Central University of Jharkhand, Ranchi, Jharkhand, India. [168]Centro de Modelación y Monitoreo de Ecosistemas, Universidad Mayor, Santiago, Chile. [169]Vicerrectoria de Investigacion y Postgrado, Universidad de La Frontera, Temuco, Chile. [170]Depto. de Silvicultura y Conservacion de la Naturaleza, Universidad de Chile, Temuco, Chile. [171]Datascientist.ch, Wallisellen, Switzerland. [172]Peoples Friendship University of Russia (RUDN University), 6 Miklukho-Maklaya St., Moscow 117198, Russia. [173]Geobotany, Faculty of Biology, University of Freiburg, Freiburg im Breisgau, Germany. [174]National Forest Centre, Forest Research Institute Zvolen, Zvolen, Slovakia. [175]Université de Lorraine, AgroParisTech, INRAE, Silva, Nancy, France. [176]Faculty of Environmental Sciences and Natural Resource Management, Norwegian University of Life Sciences, Ås, Norway. [177]Departamento de Biología, Universidad de la Serena, La Serena, Chile. [178]Centro de Ciências Biológicas e da Natureza, Universidade Federal do Acre, Rio Branco, Acre, Brazil. [179]Guyana Forestry Commission, Georgetown, French Guiana. [180]Environmental and Life Sciences, Faculty of Science, Universiti Brunei Darussalam, Bandar Seri Begawan, Brunei Darussalam. [181]Plant Systematic and Ecology Laboratory, Department of Biology, Higher Teachers' Training College, University of Yaoundé I, Yaoundé, Cameroon. [182]Departamento de Ecologia, Universidade Federal do Rio Grande do Norte, Natal, Rio Grande do Norte, Brazil. [183]Department of Forest Management, Dendrometry and Forest Economics, Warsaw University of Life Sciences, Warsaw, Poland. [184]Center for Ecological Dynamics in a Novel Biosphere (ECON-OVO), Department of Biology, Aarhus University, Ny Munkegade 114, DK-8000 Aarhus, Denmark. [185]Faculty of Forestry and Wood Sciences, Czech University of Life Sciences, Prague, Czech Republic. [186]Wildlife Conservation Society, New York, NY, USA. [187]Quantitative Biodiversity Dynamics, Dept. of Biology, Utrecht University, Utrecht, The Netherlands. [188]Iwokrama International Centre for Rain Forest Conservation and Development, Kurupukari, Guyana. [189]School of Forestry and Environmental Studies, Yale University, New Haven, CT, USA. [190]Botanical Garden of Ural Branch of Russian Academy of Sciences, Ural State Forest Engineering University, Yekaterinburg, Russia. [191]Pontificia Universidad Católica del Ecuador, Quito, Ecuador. [192]LINCGlobal, Museo Nacional de Ciencias Naturales, CSIC, Madrid, Spain. [193]Plant Ecology and Nature Conservation Group, Wageningen University, P.O. Box 47 Wageningen 6700 AA, The Netherlands. [194]Silviculture Research Institute, Vietnamese Academy of Forest Sciences, Hanoi, Vietnam. [195]Department of Biology, Stanford University, Stanford, CA, USA. [196]Q-ForestLab, Department of Environment, Ghent University, Ghent, Belgium. [197]Centre for the Research and Technology of Agro-Environmental and Biological Sciences, CITAB, University of Trás-os-Montes and Alto Douro, UTAD, Quinta de Prados 5000-801, Vila Real, Viseu, Portugal. [198]Department of Ecology and Sustainable Agriculture, Agricultural High School, Polytechnic Institute of Viseu, 3500-606 Viseu, Portugal. [199]Department of Forest Engineering Universidade Regional de Blumenau, Blumenau, Brazil. [200]Environmental Studies and Research Center, University of Campinas, UNICAMP, Campinas, Brazil. [201]Department of Forest and Wood Science, University of Stellenbosch, Stellenbosch, South Africa. [202]Key Laboratory of Tropical Biological Resources, Ministry of Education, School of Life and Pharmaceutical Sciences, Hainan University, Haikou, China. [203]Division of Forestry and Natural Resources, West Virginia University, Morgantown, WV, USA. [204]Department of Zoology, University of Oxford, Oxford, UK. [205]Manaaki Whenua–Landcare Research, Lincoln, New Zealand. [206]Department of Wetland Ecology, Institute for Geography and Geoecology, Karlsruhe Institute for Technology, Karlsruhe, Germany. [207]Independent Researcher, Sommersbergseestrasse, 8990 Bad Aussee, Austria. [208]Centre for Agricultural Research in Suriname (CELOS), Paramaribo, Suriname. [209]Tropenbos International, Wageningen, The Netherlands. [210]Polish State Forests, Coordination Center for Environmental Projects, Warsaw, Poland. [211]Research Center of Forest Management Engineering of State Forestry and Grassland Administration, Beijing Forestry University, Beijing, China.

