## [Peer Review File · Nature Communications]

Reviewers' Comments:

Reviewer #1:

Remarks to the Author:

This is a very interesting manuscript presenting elaborate analyses using to address the question whether there exist alternative stable states in leaf phenology of trees. Evidence for alternative stable states is found, driven by con-phenological positive feedbacks. Although I do find the findings generally convincing, I also have two major points of critique that prevent me from recommending publication in its current form.

The first point is related to the current framing of results. The data that are used are very much biased towards the USA and Europe. This is fine, as data limitations prevent a representative global coverage, but this does mean that these results cannot simply be extrapolated globally. In the areas where most of the data points are from, the alternative stable states would occur between broadleaf and needleleaf forests, but this would not apply everywhere. Therefore, the authors should limit their conclusions and implications to those areas where the data actually underpin them.

The second point that I would like to see addressed, ideally with an additional analysis, is about how the data on leaf phenology are treated. Species are categorized as either evergreen or deciduous. Normally this is a very reasonable distinction to make, but in this case it means that a dichotomy is created that in theory could affect the results. For example, a hypothetical tree that sheds leaves for one month per year is currently treated the same as a tree that sheds them for 11 months. However, to infer bimodality in leaf phenology, one should ideally treat leaf shedding as a spectrum, not an a priori dichotomy. Put otherwise, right now, evergreen is defined as leaf shedding = 0, and deciduous is defined as leaf shedding > 0, while leaf shedding could also be treated as a continuous variable. In that case, perhaps, the "relative abundance deciduous trees weighted by their greenness per year" could replace "relative abundance evergreen trees". Without any information about how leaf shedding is distributed within the deciduous type, one can in principle not be certain that the actual underlying distribution of leaf shedding, or deciduousness, is bimodal. To this I should add, however, that, as a resident and native of temperate Europe, I am well aware that the annual duration of leaf shedding is not distributed uniformly here, but differences along latitude may be expected.

Two more minor remarks:

L. 300-301: this suggests that soil is a "driver" of bistability, but soils are not independent of the tree types themselves. The way soils are discussed in lines 488-490 seems completely justified; please also rephrase to "soil-related positive feedbacks", or similar, in the Abstract.

L. 749: here it is stated that trees facilitate the establishment of their own phenology type. Can it be explained how that works? Wouldn't it make more sense to posit that trees facilitate their own species instead?

Reviewer #2:

Remarks to the Author:

Zou et al. present a comprehensive analysis of the presence of bimodality in forest leaf types (coniferous vs deciduous) across large spatial scales, suggesting that bimodality and hysteresis are present in forest ecosystems, and that this is largely due to soil pH driving the presence of certain species. They provide multiple lines of evidence to support this interesting and important finding.

Whilst I am not a forest ecologist, I found this paper accessible and on a very interesting topic. It is well written and appears statistically sound (although not being a forest ecologists means the intricacies of the analyses and the choices made in dealing with the forest data and the simulations is beyond my expertise to comment on).

From an alternative stable states and resilience perspective this is a clearly interesting, relevant and important piece of work with implications for how we assess the impacts of global change on biodiversity and ecosystem processes, and from this perspective I think that it clearly merits

publication.

I don't have any specific comments to make regarding clarity (in part because this is very well written, and partly because the analysis is so heavily forest focused that I cannot sensibly comment on it beyond it appears to make sense) and – assuming that the more forest focussed reviewers agree this is robust – I believe this is publishable in its current form.

Reviewer #3:

Remarks to the Author:

I've been asked to provide a technical review focused on the use of machine learning spatial mapping, and so I'll restrict my comments to this aspect of the study, which I shall note I found interesting and pleasing to read.

The authors used the random forest (RF) algorithm to map a continuous variable (noted "BI" and roughly representing the proportion of deciduous vs evergreen trees within clusters of data) at global scale based on a large number of predictors ($n = 62$). The spatial distribution of training data is strongly unbalanced, with 98% of the data located above 15 degrees northern latitude (L. 957 and Fig. 4a where we can see large clusters of data in the US and Europe), and a very scarce data distribution in the global South. In this situation (i.e. spatially clustered data, large number of predictors), broad-scale mapping tasks with RF may be challenging because (1) RF is known to overfit the spatial structure in the predictors and (2) the classical tool to assess model prediction performance (and model overfitting) i.e. the random 10-fold cross-validation (CV) provides misleading statistics. It follows that an overly optimistic assessment of model predictive performances can be made, and that model predictions (the map) may in fact be largely unreliable in large chunks of the world where there is no training data (here, the global South). These issues have been described in several papers in the last decades, and although the authors do not cite any of them, they do take certain measures to mitigate the problem: (1) as I see it, prediction maps are far from being the central part of the study's results ; (2) the only prediction map presented in the main text (Fig. 4b) is restricted to the northern hemisphere – which I think is a very good idea ; (3) the authors did try an alternative CV approach to validate one of their model (a spatially buffered leave-one-out CV, or SBLOO-CV), which I see as a gesture of goodwill. Hence, the machine learning spatial mapping component of this study, as is, does not make me go through the roof.

In the following, I provide a few advices that the authors are free to take on so that readers interested in spatial mapping, like myself, find in the manuscript (1) the necessary information to get better sense of the predictive performance of the models being presented (and of the reliability of the resulting maps) and (2) mentions to obvious limitations of this work.

(1) Provide validation statistics of purely spatial RF model

The authors provide in the main text the SBLOO-CV statistics for a buffer radius of 500 km (L. 439-441). As far as I'm aware, there is no clear-cut approach to define which buffer size might be more informative than others. Since random forest can overfit spatial structures, using spatial autocorrelation in model residuals (or its disappearance) might not be the best criteria to identify a buffer size – and the corresponding SBLOO-CV statistics – to put forward for readers. However, I may be missing something on the authors' strategy concerning model spatial model CV since there is no reference to a specific approach/study they may be following.

Instead of choosing a SBLOO-CV statistics at an arbitrary buffer size, I'd argue for an alternative (or additional) approach to provide information to reader on model predictive power. In the frame of the SBLOO-CV procedure, the authors should cross-validate both (1) their 'full' model as well as (2) a purely spatial model (i.e. a RF model based on two predictors: training data latitude and longitude). The SBLOO-CV statistics of the purely spatial model can be represented by one additional line in Fig. S12 (panels A and B).

In the main text, the authors could compare how the predictive performance of the 'full' model

change with respect to that of the purely spatial model. Ideally, the predictive power of the purely spatial model would disappear at a certain distance from training data (e.g. 500 km) while the full model would retain some predictive power. This approach would be more informative than the current one on model predictive performance, and hopefully back up the choice of such complex model (with 62 predictors) over a much simpler one (the purely spatial model).

(2) Provide validation statistics of RF model based on environmental clustering

The authors used two RF models to map BI at global scale: one is based on "spatial" clustering of data, the other one on "environmental" clustering. The "spatial" RF model is used to produce the map of the northern hemisphere in the main text (Fig. 4b). The "environmental" RF model is used to produce the global map in Fig. S16.

The "spatial" RF model has been validated by SBLOO-CV. However, the sole information readers have on the "environmental" RF model predictive performance is that the resulting map "shows consistent patterns [with the map from the "spatial" RF model] (72% agreement between models, Fig. S8 & S16)" (L. 443). It is unclear to me how this "72%" statistics was calculated. I also cannot find in the text how many Kmeans clusters were used for environmental clustering. That being said, I wonder whether this model could not be validated using the same approach as the one for the RF model based on spatial data clustering (i.e. SBLOO-CV + corresponding purely spatial model). The result of this validation could be added to Fig. S12.

Obviously, the results of such validation would be representative of model predictive performance on the northern hemisphere – while model predictive power and map reliability on the global South would still be relatively unknown given data scarcity –, but it would be better than nothing.

(3) Add some words of caution on the use of the global BI map

As eluded to in the previous comment, in my opinion the reliability of the map (for example in the Congo basin) is unknown. Over the global South, models are in a situation of predictive extrapolation, which we can decompose into geographic and environmental extrapolations.

The geographic extrapolation component in the Global South is virtually impossible to assess given the scarcity of the data in this part of the world (notably in the two main basins of tropical dense forests). While it could be argued that the SBLOO-CV – that the authors used – is designed to assess this uncertainty component, the results presented in the study are obviously steered by / representative of the northern hemisphere (where data are located). Whether they are representative of model performance on tropical forest systems is uncertain.

Concerning the environmental extrapolation component, the authors performed a fairly liberal assessment whereby (1) a PCA was performed on the 62 predictors at training data location, (2) convex hulls were built for each bivariate combinations of the top 22 PCA axes and (3) map pixels were classified as being inside or outside the predictor space when they fall within or outside of the hulls. Ideally, a unique convex hull should have been built on all 62 predictors (or all top 22 PCA axes), not on bivariate combinations. This would probably be extremely computer-demanding (hence the bivariate combinations approach), but the results would be much more realistic. It is indeed easy to conceive that a map pixel may fall within two bivariate sets of environmental gradients covered by training data, but outside the hull formed by the four gradients taken together. The more gradients are considered (22 here), the more optimistic the "bivariate approach" is. For this reason, as a reader, I have little confidence in the result of the interpolation vs extrapolation assessment presented in Fig. S9c.

Last, the authors present a map of predictions standard-deviation across bootstrap iterations as an assessment of model uncertainty (Fig. S9b). The standard-deviation of model predictions across bootstrap iterations merely quantify the "stability" of model predictions. It measures one uncertainty component affecting model "precision" (not "accuracy", as stated in the figure caption, L. 348), but do not account for a bunch of other factors that may contribute to model uncertainty (e.g., tree species misidentification, wrong assignment of leaf habits, uncertainty in covariate values, to cite only a few examples). The resulting map – while being quite difficult to interpret –

is pretty telling, with nearly no prediction uncertainty on tropical dense forest basins. This could lead uninformed reader to think that the map is perfectly reliable in the tropics (and perhaps use the map in their own research), which I don't think is what the authors intend to do.

For these reasons (difficulties to assess geographic extrapolation, liberal assessment of environmental extrapolation, weak assessment of model uncertainty), I'd advice the author to add a word of caution on the use of this map in the "limitation" paragraph of the discussion section. Further, to me, the unbalanced spatial distribution of data at world scale, and the sheer lack of data in the global South, is the primary limitation of this study, yet it is not mentioned in the discussion.

Reviewer #4:

Remarks to the Author:

In their paper "Positive feedbacks and alternative stable states in forest leaf types", Zou et al analyzed alternative stable states in the presence of evergreen and deciduous forest types and provided a global map of forest type patterns based on a random forest modelling approach. In this review I will mainly focus on the use of machine learning for spatial mapping.

Assumption of unmanaged and undisturbed areas

The assumption for the reference data is, that they represent unmanaged and undisturbed forest sites. For the majority of the available data, no information is available whether this holds true or not. To account for that, data were filtered to remove monoculture (1 species only). I don't think that this is sufficient. At least for temperate Europe, the majority of the forests have to be considered as being highly managed and patches of either broadleaf or evergreen forest are mainly a result of management decisions. In other areas, like in Siberia, fire ecology plays a major role and broadleaf forest often comes up after fire disturbance and is replaced by evergreen forest after several years as the natural succession. Therefore, I'm not convinced that the data can be used to answer the research questions of this study.

Purpose of the global predictions

The authors use the reference data to train a random forest model based on a large number of environmental variables to make predictions of forest type patterns for the entire forest area of the northern hemisphere. However, the predictions have to be understood as the expected forest type based on the environmental conditions. I wonder why such a model is necessary because high resolution data of forest types exist or could be produced based on remote sensing data. E.g. for Europe a satellite-based 10m resolution dataset of the dominant leaf type exist (<https://land.copernicus.eu/en/products/high-resolution-layer-dominant-leaf-type>) and I assume there are similar datasets for other areas or could be produced based on satellite imagery. Certainly the remote sensing based mapping will have limitations as well, but it would not suffer from problems of extrapolation as it does here due to the limited availability of reference data (see comments below). From my perspective this would be more suitable to derive the bimodality index and study the occurrence of stable states.

Suitability of the predictors

One of the main predictors of leaf types are soil properties. The data used here are predictions as well, which were made based on a similar approach as presented here and with a similar lack of data in certain areas. Therefore, the soil dataset likely misses out unique locations and as a consequence, the error propagates here because small-scale variation in soil properties might not be detected. Small patches of other soil types for example, may lead to different leaf types which might here falsely be interpreted as alternative stable states because the environment appears to be the same on the basis of the predictor data.

To acknowledge this issue, the authors compared the soil grids with data from WoSIS (Fig. S22). The locations available for comparison, however, are again limited. Also, as far as I'm aware of, the WoSIS data are used to create the global soil grids, therefore they are not suitable to indicate the suitability of the soil grids.

I recommend limiting predictions to geographic areas where the predictors are not affected by

severe extrapolation.

Spatial extrapolation

I highly acknowledge the effort to account for extrapolation conditions which was done here based on convex hulls in the feature space. However, the data point density is not considered in this approach. Some areas might only be covered by a single data point alone (e.g. Siberia is poorly covered). Again, I recommend limiting predictions to geographic areas where the predictors are not affected by severe extrapolation. This may involve extrapolation in feature but also in geographic space. I highly acknowledge Fig. S12. However, the figure shows that the performance considerably decreases with distance. When data in the proximity of 1000km are left out, the model performance is rather low. This might be an indicator for a low performance in areas that are poorly covered by data – which is not reflected by the given R^2 performance value.

Differentiated analysis of the model performance

The prediction model had a performance indicated by an R^2 of 0.52 (I take the spatial CV results here as a reference because the random approach is of limited value here). I would like to see a more differentiated view on this, e.g. on the performance in different regions. I also miss a scatter-plot showing predicted versus observed (based on the spatially held back data!) to make sure that the acceptable R^2 is not just due to a high ability of the model to differentiate between the two extremes while having a low ability to predict the intermediate values of the BI. I'm surprised about the high RMSE values of approx. 0.4 (Fig S12). Given the BI range from -1 to 1, this is a lot and questions the conclusions in general. Again, I would like to see visually how predicted and observed values compare (based on the spatial CV) to get more insights into the performance.

Summary

I would be very careful with drawing conclusions from the results due to the concerns mentioned above.

RESPONSE TO REVIEWER 1

[Comment 1]

This is a very interesting manuscript presenting elaborate analyses using to address the question whether there exist alternative stable states in leaf phenology of trees. Evidence for alternative stable states is found, driven by con-phenological positive feedbacks. Although I do find the findings generally convincing, I also have two major points of critique that prevent me from recommending publication in its current form.

Response 1: We thank the reviewer for the positive and constructive feedback on our paper. We also very much appreciate the concerns raised by the reviewer and believe we were able to address all of them as outlined below.

[Comment 2]

The first point is related to the current framing of results. The data that are used are very much biased towards the USA and Europe. This is fine, as data limitations prevent a representative global coverage, but this does mean that these results cannot simply be extrapolated globally. In the areas where most of the data points are from, the alternative stable states would occur between broadleaf and needleleaf forests, but this would not apply everywhere. Therefore, the authors should limit their conclusions and implications to those areas where the data actually underpin them.

Response 2: We entirely agree with this take and have now extended the discussion to emphasize the focus on northern latitudes and regions with low extrapolation risks (L.513-519): “Finally, our analysis of phenological bimodality carries a heightened risk of extrapolation errors when applied to tropical regions and the Global South, attributed primarily to a significant data deficiency. As a result, we have confined our predictions and conclusions to forest regions 1) located above 15 degrees northern latitude, 2) whose environmental conditions fully represented by our forest plot data. This decision is underpinned by the fact that over 98% of the data in the GFBI database, which our spatial model relies on, originates from these temperate zones of the Northern Hemisphere. This geographical delimitation ensures that our findings are based on a robust and representative dataset, minimizing the risks associated with overextending our conclusions to poorly represented regions.”

[Comment 3]

The second point that I would like to see addressed, ideally with an additional analysis, is about how the data on leaf phenology are treated. Species are categorized as either evergreen or deciduous. Normally this is a very reasonable distinction to make, but in this case it means that a dichotomy is created that in theory could affect the results.

For example, a hypothetical tree that sheds leaves for one month per year is currently treated the same as a tree that sheds them for 11 months. However, to infer bimodality in leaf phenology, one should ideally treat leaf shedding as a spectrum, not an a priori dichotomy. Put otherwise, right now, evergreen is defined as leaf shedding = 0, and deciduous is defined as leaf shedding > 0, while leaf shedding could also be treated as a continuous variable. In that case, perhaps, the “relative abundance deciduous trees weighted by their greenness per year” could replace “relative abundance evergreen trees”. Without any information about how leaf shedding is distributed within the deciduous type, one can in principle not be certain that the actual underlying distribution of leaf shedding, or deciduousness, is bimodal. To this I should add, however, that, as a resident and native of temperate Europe, I am well aware that the annual duration of leaf shedding is not distributed uniformly here, but differences along latitude may be expected.

Response 3: Thank you for your constructive feedback on our treatment of leaf phenology within our study. We recognize the potential limitations of categorizing species strictly as either evergreen or deciduous, a dichotomy that simplifies the complex spectrum of leaf shedding behaviors across different species.

To test this, we examined whether this bimodality trend is apparent if we treat deciduous as a continuous or categorical trait. In this case we found that, whether deciduousness is treated as a category or a continuous trait does not affect the bimodal distribution between evergreen and deciduous species. As such, it has no bearing on the overall conclusions. To demonstrate that the actual underlying distribution of leaf shedding, or deciduousness, is bimodal, we now used the most comprehensive species-level dataset on growing season length from Zohner et al. (2017), which includes data for more than 400 species from North America, Europe and Asia¹. The attached figure shows that across all continents, there is a clear bimodal distribution, which was also statistically confirmed by the significant P values of a Hartigan’s dip test.

Moreover, it is important to clarify that our study's focus is on regions with pronounced seasonal variations (latitudes greater than 15° N), where the distinction between deciduous and evergreen species is not only clear but also rooted in their genetic

makeup. For example, deciduous trees, such as beeches, invariably lose their leaves in response to winter conditions, while evergreens, like spruces, retain their foliage throughout the year. This genetically determined behavior results in predictable patterns of leaf phenology that are consistent across individuals of the same species, despite spatial variations in the length of the leafless season.

Our study's scope is specifically tailored to understand these genetically fixed phenological patterns within a clear seasonal framework. Including additional variables, such as the precise length of the growing season for each species, while potentially enriching in a different context, would not alter the classification of a species as deciduous or evergreen within the studied latitudes. This is because, in the regions we are examining, an evergreen tree's inherent genetic traits ensure its evergreen status, and similarly, a deciduous tree remains deciduous regardless of slight variations in the growing season's length.

Our decision to adhere to this dichotomous classification system is driven by the genetic demarcation between deciduous and evergreen species in these latitudes, coupled with the practical limitations of data availability. Extending our analysis to account for the nuanced length of the growing season across the vast geographic and climatic spectrum we are studying would necessitate a level of detailed phenological data that is currently beyond our reach. Adding information on the precise growing season length of each individual in each plot is simply not possible. Species-level information would not add much to this question, as there is too much geographic variation within species, which is often higher than the variation across species.

However, we acknowledge the complexity of leaf phenology in tropical regions, where the delineation between evergreen and deciduous species is more fluid and predominantly influenced by environmental conditions such as drought. Here, the same species may exhibit evergreen characteristics in consistently moist environments and shift towards deciduousness in areas experiencing a distinct dry season. This observation aligns with our spatial analysis, indicating that the phenomenon of alternative stable states is comparatively less common in tropical settings. Unlike temperate and boreal regions, where leaf shedding and retention are genetically determined, tropical regions present a scenario where deciduousness is more directly shaped by external environmental factors.

In summary, while we share the reviewer's call for a more detailed analysis that captures the spectrum of leaf shedding, the scope of our analysis and current limitations of available data necessitates the use of a more generalized approach. We appreciate the suggestion and acknowledge the potential for future studies to build upon our work, incorporating a more nuanced view of leaf phenology as data availability improves.

[Comment 4]

L. 300-301: this suggests that soil is a “driver” of bistability, but soils are not independent of the tree types themselves. The way soils are discussed in lines 488-490 seems completely justified; please also rephrase to “soil-related positive feedbacks”, or similar, in the Abstract.

Response 4: We have rephrased the sentence in the abstract (L.301) from “which are likely driven by soil conditions” to “which are likely driven by soil-related positive feedbacks”.

[Comment 5]

L. 749: here it is stated that trees facilitate the establishment of their own phenology type. Can it be explained how that works? Wouldn't it make more sense to posit that trees facilitate their own species instead?

Response 5: While there may of course be facilitation at the species level, we want to emphasize here that such facilitation may also act at the functional type level. There have been a few recent studies to suggest that the facilitation can happen at the scale of functional type. For example, such a distinction has recently been shown to exist between ectomycorrhizal trees and arbuscular mycorrhizal trees, each of which are comprised of many species (Averill et al. 2022). We provide a possible explanation for this in L.322-329:

“However, evidence suggests that biological feedbacks within forest stands may also drive leaf phenology strategies, whereby the dominant type in a community may favor the establishment and survival of its own type (the con-phenological feedback). For example, many evergreen trees (especially conifers) have an advantage over deciduous species in nutrient-poor and acidic soils. High concentrations of tannins and phenols as well as low N concentration in evergreen leaves decrease the soil pH and rates of leaf decomposition, further limiting soil fertility and, in turn, favoring the dominance of evergreen species. Similarly, deciduous trees may also favor their own phenology type by shedding less tannic, nutrient-rich leaves that can quickly be decomposed. Furthermore, the dominance of either leaf type can lead to an accumulation of ‘con-phenological’ seeds and seedlings, which may strengthen the positive feedback.”.

RESPONSE TO REVIEWER 2

[Comment 1]

Zou et al. present a comprehensive analysis of the presence of bimodality in forest leaf types (coniferous vs deciduous) across large spatial scales, suggesting that bimodality and hysteresis are present in forest ecosystems, and that this is largely due to soil pH driving the presence of certain species. They provide multiple lines of evidence to support this interesting and important finding.

Whilst I am not a forest ecologist, I found this paper accessible and on a very interesting topic. It is well written and appears statistically sound (although not being a forest ecologist means the intricacies of the analyses and the choices made in dealing with the forest data and the simulations is beyond my expertise to comment on).

From an alternative stable states and resilience perspective this is a clearly interesting, relevant and important piece of work with implications for how we assess the impacts of global change on biodiversity and ecosystem processes, and from this perspective I think that it clearly merits publication.

I don't have any specific comments to make regarding clarity (in part because this is very well written, and partly because the analysis is so heavily forest focused that I cannot sensibly comment on it beyond it appears to make sense) and – assuming that the more forest focussed reviewers agree this is robust – I believe this is publishable in its current form.

Response 6: We thank the reviewer for the very positive feedback, and for pointing out the importance of our study. We are very pleased to hear that reviewer 2 sees our work as a compelling contribution to Nature Communications.

RESPONSE TO REVIEWER 3

[Comment 1]

I've been asked to provide a technical review focused on the use of machine learning spatial mapping, and so I'll restrict my comments to this aspect of the study, which I shall note I found interesting and pleasing to read.

Response 7: We thank the reviewer for the positive and constructive feedback on our paper.

[Comment 2]

The authors used the random forest (RF) algorithm to map a continuous variable (noted "BI" and roughly representing the proportion of deciduous vs evergreen trees within clusters of data) at global scale based on a large number of predictors ($n = 62$). The spatial distribution of training data is strongly unbalanced, with 98% of the data located above 15 degrees northern latitude (L. 957 and Fig. 4a where we can

see large clusters of data in the US and Europe), and a very scarce data distribution in the global South. In this situation (i.e. spatially clustered data, large number of predictors), broad-scale mapping tasks with RF may be challenging because (1) RF is known to overfit the spatial structure in the predictors and (2) the classical tool to assess model prediction performance (and model overfitting) i.e. the random 10-fold cross-validation (CV) provides misleading statistics. It follows that an overly optimistic assessment of model predictive performances can be made, and that model predictions (the map) may in fact be largely unreliable in large chunks of the world where there is no training data (here, the global South). These issues have been described in several papers in the last decades, and although the authors do not cite any of them, they do take certain measures to mitigate the problem: (1) as I see it, prediction maps are far from being the central part of the study's results ; (2) the only prediction map presented in the main text (Fig. 4b) is restricted to the northern hemisphere – which I think is a very good idea ; (3) the authors did try an alternative CV approach to validate one of their model (a spatially buffered leave-one-out CV, or SBLOO-CV), which I see as a gesture of goodwill. Hence, the machine learning spatial mapping component of this study, as is, does not make me go through the roof.

In the following, I provide a few advices that the authors are free to take on so that readers interested in spatial mapping, like myself, find in the manuscript (1) the necessary information to get better sense of the predictive performance of the models being presented (and of the reliability of the resulting maps) and (2) mentions to obvious limitations of this work.

Response 8: We greatly appreciate the concerns raised by the reviewer and believe we were able to address all of them as outlined below.

[Comment 3]

(1) Provide validation statistics of purely spatial RF model

The authors provide in the main text the SBLOO-CV statistics for a buffer radius of 500 km (L. 439-441). As far as I'm aware, there is no clear-cut approach to define which buffer size might be more informative than others. Since random forest can overfit spatial structures, using spatial autocorrelation in model residuals (or its disappearance) might not be the best criteria to identify a buffer size – and the corresponding SBLOO-CV statistics – to put forward for readers. However, I may be missing something on the authors' strategy concerning model spatial model CV since there is no reference to a specific approach/study they may be following.

Instead of choosing a SBLOO-CV statistics at an arbitrary buffer size, I'd argue for an alternative (or additional) approach to provide information to reader on model predictive power. In the frame of the SBLOO-CV procedure, the authors should cross-validate both (1) their 'full' model as well as (2) a purely spatial model (i.e. a RF model based on two predictors: training data latitude and longitude). The SBLOO-CV

statistics of the purely spatial model can be represented by one additional line in Fig. S12 (panels A and B).

In the main text, the authors could compare how the predictive performance of the 'full' model change with respect to that of the purely spatial model. Ideally, the predictive power of the purely spatial model would disappear at a certain distance from training data (e.g. 500 km) while the full model would retain some predictive power. This approach would be more informative than the current one on model predictive performance, and hopefully back up the choice of such complex model (with 62 predictors) over a much simpler one (the purely spatial model).

Response 9: We thank the reviewer for this suggestion. We have now trained a purely spatial model (null model) and compared it with our full model in the updated Figure S12. The result shows that under any buffer radius, the full model has better predictive performance than the null model. And the predictive power of the null model disappears at a radius of around 1000 km. We have updated the Supporting Information accordingly in L.84-85.

[Comment 4]

(2) Provide validation statistics of RF model based on environmental clustering

The authors used two RF models to map BI at global scale: one is based on "spatial" clustering of data, the other one on "environmental" clustering. The "spatial" RF model is used to produce the map of the northern hemisphere in the main text (Fig. 4b). The "environmental" RF model is used to produce the global map in Fig. S16.

The "spatial" RF model has been validated by SBLOO-CV. However, the sole information readers have on the "environmental" RF model predictive performance is that the resulting map "shows consistent patterns [with the map from the "spatial" RF model] (72% agreement between models, Fig. S8 & S16)" (L. 443). It is unclear to me how this "72%" statistics was calculated. I also cannot find in the text how many Kmeans clusters were used for environmental clustering. That being said, I wonder whether this model could not be validated using the same approach as the one for the RF model based on spatial data clustering (i.e. SBLOO-CV + corresponding purely spatial model). The result of this validation could be added to Fig. S12.W

Obviously, the results of such validation would be representative of model predictive performance on the northern hemisphere – while model predictive power and map reliability on the global South would still be relatively unknown given data scarcity –, but it would be better than nothing.

Response 10: We thank the reviewer for raising this point. We made editions in L. 733-742 of the Method part: "In the second partitioning method ("environmental clustering" approach), we implemented K-means clustering to group forest plots into 15000 clusters (thus close to the sample size of spatial clustering) based on the leading 3 environmental PCs of each plot. We removed clusters with fewer than 10 forest plots,

and finally got 14858 clusters. Similarly, for each cluster, we aggregated plot-level relative evergreen abundance to calculate the BI.” “The environmental clustering approach ensures that each cluster consists of environmentally homogenous plots, while plots within clusters are not necessarily in spatial proximity.” The 72% agreement referred to an R^2 of 0.72 when correlating the predictions from the spatial RF against the predictions from the environmental RF (now clarified in L. 739-741). The environmental clustering method didn’t use any spatial information but only used environmental conditions (also see Fig. S8), therefore can’t be validated via spatial cross-validation. To avoid extrapolation in the global south, we now limit our predictions (see updated Fig. S16) to the Northern hemisphere as in Fig. 4B.

[Comment 5]

(3) Add some words of caution on the use of the global BI map

As eluded to in the previous comment, in my opinion the reliability of the map (for example in the Congo basin) is unknown. Over the global South, models are in a situation of predictive extrapolation, which we can decompose into geographic and environmental extrapolations.

The geographic extrapolation component in the Global South is virtually impossible to assess given the scarcity of the data in this part of the world (notably in the two main basins of tropical dense forests). While it could be argued that the SBLOO-CV – that the authors used – is designed to assess this uncertainty component, the results presented in the study are obviously steered by / representative of the northern hemisphere (where data are located). Whether they are representative of model performance on tropical forest systems is uncertain.

Response 11: We thank the reviewer for this suggestion. We have added a caution in L. 513-519 of the discussion accordingly: “Finally, our analysis of phenological bimodality carries a heightened risk of extrapolation errors when applied to tropical regions and the Global South, attributed primarily to a significant data deficiency. As a result, we have confined our predictions and conclusions to forest regions 1) located above 15 degrees northern latitude, 2) whose environmental conditions fully represented by our forest plot data. This decision is underpinned by the fact that over 98% of the data in the GFBI database, which our spatial model relies on, originates from these temperate zones of the Northern Hemisphere. This geographical delimitation ensures that our findings are based on a robust and representative dataset, minimizing the risks associated with overextending our conclusions to poorly represented regions.”.

[Comment 6]

Concerning the environmental extrapolation component, the authors performed a fairly liberal assessment whereby (1) a PCA was performed on the 62 predictors at

training data location, (2) convex hulls were built for each bivariate combinations of the top 22 PCA axes and (3) map pixels were classified as being inside or outside the predictor space when they fall within or outside of the hulls. Ideally, a unique convex hull should have been built on all 62 predictors (or all top 22 PCA axes), not on bivariate combinations. This would probably be extremely computer-demanding (hence the bivariate combinations approach), but the results would be much more realistic. It is indeed easy to conceive that a map pixel may fall within two bivariate sets of environmental gradients covered by training data, but outside the hull formed by the four gradients taken together. The more gradients are considered (22 here), the more optimistic the “bivariate approach” is. For this reason, as a reader, I have little confidence in the result of the interpolation vs extrapolation assessment presented in Fig. S9c.

Response 12: We are grateful for the reviewer's constructive suggestion. In alignment with this, we acknowledge that incorporating all top 22 Principal Component Analysis (PCA) axes to construct the convex hull would indeed provide a more comprehensive representation. However, as correctly identified by the reviewer, we encountered significant computational limitations.

We now successfully generated convex hulls using the top two, three, and four PCA axes, which are illustrated in the interpolation versus extrapolation maps in Figure S31. Unfortunately, when attempting to expand this to five or more dimensions, both our local computing resources and the ETH Euler Cluster were unable to handle the memory and computational time required for such extensive calculations.

To address this challenge, we now additionally employed an alternative approach. For each pixel in our analysis, we calculated the proportion of the 62 predictors whose values fell within the range of our training dataset. This method yielded a map that displays the percentage of interpolated predictors, now updated in Figure S9A. We have also included a detailed description of this methodology in the Supporting Information (L.71-76). This additional analysis serves as a straightforward complementary method to our primary PCA-based approach, helping to mitigate the constraints posed by our computational resources.

[Comment 7]

Last, the authors present a map of predictions standard-deviation across bootstrap iterations as an assessment of model uncertainty (Fig. S9b). The standard-deviation of model predictions across bootstrap iterations merely quantify the “stability” of model predictions. It measures one uncertainty component affecting model “precision” (not “accuracy”, as stated in the figure caption, L. 348), but do not account for a bunch of other factors that may contribute to model uncertainty (e.g., tree species misidentification, wrong assignment of leaf habits, uncertainty in covariate values, to cite only a few examples). The resulting map – while being quite difficult to

interpret – is pretty telling, with nearly no prediction uncertainty on tropical dense forest basins. This could lead uninformed reader to think that the map is perfectly reliable in the tropics (and perhaps use the map in their own research), which I don't think is what the authors intend to do.

Response 13: We thank the reviewer for pointing this out. We have deleted this figure and replaced it with the new interpolation maps (see updated Fig. S9).

[Comment 8]

For these reasons (difficulties to assess geographic extrapolation, liberal assessment of environmental extrapolation, weak assessment of model uncertainty), I'd advice the author to add a word of caution on the use of this map in the "limitation" paragraph of the discussion section. Further, to me, the unbalanced spatial distribution of data at world scale, and the sheer lack of data in the global South, is the primary limitation of this study, yet it is not mentioned in the discussion.

Response 14: We thank the reviewer for this comment. Please check Response 11 and we have added this caution to the end of the discussion.

RESPONSE TO REVIEWER 4

[Comment 1]

In their paper "Positive feedbacks and alternative stable states in forest leaf types", Zou et al analyzed alternative stable states in the presence of evergreen and deciduous forest types and provided a global map of forest type patterns based on a random forest modelling approach.

In this review I will mainly focus on the use of machine learning for spatial mapping.

Assumption of unmanaged and undisturbed areas

The assumption for the reference data is, that they represent unmanaged and undisturbed forest sites. For the majority of the available data, no information is available whether this holds true or not. To account for that, data were filtered to remove monoculture (1 species only). I don't think that this is sufficient. At least for temperate Europe, the majority of the forests have to be considered as being highly managed and patches of either broadleaf or evergreen forest are mainly a result of management decisions. In other areas, like in Siberia, fire ecology plays a major role and broadleaf forest often comes up after fire disturbance and is replaced by evergreen forest after several years as the natural succession. Therefore, I'm not convinced that the data can be used to answer the research questions of this study.

Response 15: We gratefully acknowledge the reviewer for highlighting the potential impact of human management and disturbance on forest composition. We have added

a caution of the potential human impact in L.495-497 of the discussion. We have taken specific measures to account for these factors in our datasets. For the FIA dataset, the major dataset used for proving bimodality and positive feedback (Fig.1A-C, Fig. 2 and Fig. 3), we possess detailed information on forest management practices. Consequently, we excluded plots identified as plantations or those undergoing active management or harvesting.

Regarding the Global Forest Biodiversity Initiative (GFBi) and FunDivEUROPE datasets, management data is available for only a subset of plots. To mitigate the influence of human management, particularly monoculture plantations (which numerous studies^{2,3} have identified as the predominant form of human-induced forest composition change), we employed rigorous plot exclusion criteria. Specifically, we excluded plots: 1) with fewer than ten tree individuals, 2) containing only a single species, and 3) where the relative basal area of any species exceeded 75% of the total cumulative basal area (L.546-547). While we acknowledge this method may not completely eliminate human influence, it represents a robust approach to minimize its impact, given the limitations of the available data. This challenge of potential human influence is not unique to our study but is a common issue encountered in high-level research utilizing these datasets^{4,5}.

Regarding the aspect of succession, we have conducted an ancillary analysis focusing solely on late-successional forests, paralleling the approach in Figures 1 and 2. Given the lack of comprehensive forest age data across our plots, we used the plot-mean diameter at breast height (DBH) as a surrogate for forest successional stage and age, based on established research^{6,7}. We calculated the plot-mean DBH for all GFBi plots and determined the 0.75 quantile threshold (approximately 25 cm) to delineate late-successional plots. We then filtered the GFBi, FIA, and FunDivEUROPE datasets to include only plots exceeding this DBH threshold. The subsequent bimodality and demographic analyses across these refined datasets (Fig. S20) corroborate our initial findings (Figs. 1-2), reinforcing the robustness of our conclusions. The methodologies for these additional analyses are detailed in lines 99-117 of the Supporting Information.

[Comment 2]

Purpose of the global predictions

The authors use the reference data to train a random forest model based on a large number of environmental variables to make predictions of forest type patterns for the entire forest area of the northern hemisphere. However, the predictions have to be understood as the expected forest type based on the environmental conditions. I wonder why such a model is necessary because high resolution data of forest types exist or could be produced based on remote sensing data. E.g. for Europe a satellite-based 10m resolution dataset of the dominant leaf type exist (<https://land.copernicus.eu/en/products/high-resolution-layer-dominant-leaf-type>) and I assume there are similar datasets for other areas or could be produced based on

satellite imagery. Certainly the remote sensing based mapping will have limitations as well, but it would not suffer from problems of extrapolation as it does here due to the limited availability of reference data (see comments below). From my perspective this would be more suitable to derive the bimodality index and study the occurrence of stable states.

Response 16: We are grateful to the reviewer for highlighting the need for clarity in our approach to predictive mapping. Our decision to create the predictive maps was guided by the following considerations:

1. **Data Resolution and Coverage Limitations:** The highest resolution dataset available for leaf type, which is at 10 meters and covers only Europe, significantly restricts our global analysis. The best alternative global dataset we found is from the European Space Agency (ESA) (<https://cds.climate.copernicus.eu/cdsapp#!/dataset/10.24381/cds.006f2c9a?tab=overview>), but it offers a resolution of only 300 meters. Such coarse resolution is insufficient for our needs, as our analysis of the Bimodality Index (BI) requires detailed information on individual trees within each 1-hectare plot to calculate the plot-level relative evergreen abundance (relEV), and subsequently to assess the distribution of relEV across plots within each 20km x 20km grid.
2. **Dataset Specificity Issues:** The 10m resolution dataset for Europe differentiates only between coniferous and broadleaf forests, not explicitly between evergreen and deciduous types. While there is a strong correlation between needle-leaf form and evergreen habit in European temperate forests (and similarly for broadleaf and deciduous), this dataset cannot be used for mapping bimodality between evergreen and deciduous forests due to the presence of broadleaf evergreens in regions like Spain, Italy, and France, as well as coniferous deciduous species in central Europe.

To align our model predictions with this 10 m resolution remote sensing data, we adopted an approximate approach: treating coniferous as evergreen and broadleaf as deciduous, and considering each pixel as one canopy. We then computed the percentage of coniferous pixels within each 100m x 100m grid and calculated BI for all such grids within each 20km x 20km grid. The resulting scatter plot between remote sensing observations and our model predictions yielded an R^2 of 0.35 (Fig. S32A), indicating reasonable agreement, particularly in bimodal and evergreen-dominated regions. However, we observed that our model predicts more evergreen dominance for “deciduous-dominated” forests shown in remote sensing (top left corner of Fig. S32A), likely stemming from our approximation of broadleaf as deciduous for the 10m resolution dataset, thus overlooking broadleaf-evergreen forests.

To address this, we used the 300m resolution forest type map to refine the 10m resolution dataset, filtering it to retain only needleleaf-evergreen, broadleaf-deciduous, and mixed forests. Applying the same comparative procedure with this refined dataset

showed improved correlation ($R^2=0.44$, Fig. S32B), further validating our model predictions and highlighting the limitations of the 10m resolution EU dataset's categorization of forests.

These methodological details, including the rationale and implementation of our approaches, are elaborated in lines 243-257 of the Supporting Information. This comprehensive explanation underscores our effort to balance the limitations of available datasets with the need for accurate and relevant predictive mapping in our study.

[Comment 3]

Suitability of the predictors

One of the main predictors of leaf types are soil properties. The data used here are predictions as well, which were made based on a similar approach as presented here and with a similar lack of data in certain areas. Therefore, the soil dataset likely misses out unique locations and as a consequence, the error propagates here because small-scale variation in soil properties might not be detected. Small patches of other soil types for example, may lead to different leaf types which might here falsely be interpreted as alternative stable states because the environment appears to be the same on the basis of the predictor data.

To acknowledge this issue, the authors compared the soil grids with data from WoSIS (Fig. S22). The locations available for comparison, however, are again limited. Also, as far as I'm aware of, the WoSIS data are used to create the global soil grids, therefore they are not suitable to indicate the suitability of the soil grids.

I recommend limiting predictions to geographic areas where the predictors are not affected by severe extrapolation.

Response 17: We thank the reviewer for pointing this out. The main reason for using WoSIS was to include plot-level observations of soil characteristics and to test if these confirm the conclusions from the Soil Grid spatial model (Fig. 5B), namely that soil chemical variables better explain variation in leaf phenology than climate within bistable forest clusters (Fig. S23-24). The reviewer is absolutely correct that WoSIS was used to create Soil Grids. However, the soil grids predictions were extrapolated using climate variables and the spatial predictions are thus likely to contain significant climate signal, while the plot-level WoSIS observations are raw measurements that are entirely independent of such spatial smoothing.

However, we very much appreciate the suggestion of limiting the analysis to areas that are not affected by extrapolation, and instead to focus on the areas that fall within the range of the data. To limit predictions to geographic areas where the predictors are not affected by severe extrapolation, we now excluded regions 1) below 15° northern latitude (represented by only < 2% of GFBI data), 2) with environmental conditions

falling outside our training data. This is now also discussed in L.513-519 of the discussion.

[Comment 4]

Spatial extrapolation

I highly acknowledge the effort to account for extrapolation conditions which was done here based on convex hulls in the feature space. However, the data point density is not considered in this approach. Some areas might only be covered by a single data point alone (e.g. Siberia is poorly covered). Again, I recommend limiting predictions to geographic areas where the predictors are not affected by severe extrapolation. This may involve extrapolation in feature but also in geographic space. I highly acknowledge Fig. S12. However, the figure shows that the performance considerably decreases with distance. When data in the proximity of 1000km are left out, the model performance is rather low. This might be an indicator for a low performance in areas that are poorly covered by data – which is not reflected by the given R^2 performance value.

Response 18: These are great points. See response 12 for a detailed description of how we revised the extrapolation analysis. In addition, we now only made predictions for forest regions (see updated Fig. 4) 1) above 15 degrees northern latitude, where > 98% of the GFBi data are located, 2) whose environmental conditions well represented by our training data (> 90% interpolation, see Fig. S9A). For the second point, we have trained a purely spatial model (null model) as a reference and compared it with our full model in the updated Figure S12. The result shows that under any buffer radius, the full model has better predictive performance than the null model. And the predictive power of the null model disappears at a radius of around 1000km (Fig. S12 and Fig. S29G), while the full model still has fair performance (with an R^2 of 0.33, Fig. S12 and Fig. S28G).

[Comment 5]

Differentiated analysis of the model performance

The prediction model had a performance indicated by an R^2 of 0.52 (I take the spatial CV results here as a reference because the random approach is of limited value here). I would like to see a more differentiated view on this, e.g. on the performance in different regions. I also miss a scatter-plot showing predicted versus observed (based on the spatially held back data!) to make sure that the acceptable R^2 is not just due to a high ability of the model to differentiate between the two extremes while having a low ability to predict the intermediate values of the BI.

I'm surprised about the high RMSE values of approx. 0.4 (Fig S12). Given the BI range from -1 to 1, this is a lot and questions the conclusions in general. Again, I would like to see visually how predicted and observed values compare (based on the spatial CV) to get more insights into the performance.

Response 19: We thank the reviewer for these suggestions. We have added new analyses to address these issues. Firstly, we checked the spatial CV R^2 for both the full model and the null model (purely spatial) within different biomes (Fig. S30). Across all biomes and buffer sizes, the full model outperformed the null model. In addition, models performed the best for the temperate biome, where most of our training data came from. Secondly, we now show scatter plots of predicted versus observed values for each buffer size for both the full and null models. The full model has fairly good performance in predicting bimodal regions, and always performs better than the null models under any radius (Fig. S28-29).

- 1 Zohner, C. M. & Renner, S. S. Innately shorter vegetation periods in North American species explain native–non-native phenological asymmetries. *Nature Ecology & Evolution* **1**, 1655-1660, doi:10.1038/s41559-017-0307-3 (2017).
- 2 Liu, C. L. C., Kuchma, O. & Krutovsky, K. V. Mixed-species versus monocultures in plantation forestry: Development, benefits, ecosystem services and perspectives for the future. *Global Ecology and Conservation* **15**, e00419, doi:<https://doi.org/10.1016/j.gecco.2018.e00419> (2018).
- 3 Du, Z. *et al.* A global map of planting years of plantations. *Scientific Data* **9**, 141, doi:10.1038/s41597-022-01260-2 (2022).
- 4 Liang, J. *et al.* Positive biodiversity-productivity relationship predominant in global forests. *Science* **354**, aaf8957, doi:10.1126/science.aaf8957 (2016).
- 5 Delavaux, C. S. *et al.* Native diversity buffers against severity of non-native tree invasions. *Nature* **621**, 773-781, doi:10.1038/s41586-023-06440-7 (2023).
- 6 Bruelheide, H. *et al.* Community assembly during secondary forest succession in a Chinese subtropical forest. *Ecol Monogr* **81**, 25-41, doi:<https://doi.org/10.1890/09-2172.1> (2011).
- 7 Buchholz, K. & Pickering, J. L. DBH-Distribution Analysis: An Alternative to Stand-Age Analysis. *Bulletin of the Torrey Botanical Club* **105**, 282-288, doi:10.2307/2484921 (1978).

Reviewers' Comments:

Reviewer #1:

First of all, I thank the authors for their responses. In my previous review, I raised two major points and several minor points. Although the minor points have been implemented to my satisfaction, I am sorry to say that the authors did not treat my major points as such.

My first major comment referred to the fact that the analyses have limited geographic scope. Data that were analysed are mostly restricted to temperate North America and Europe, which, as I explained, is not a problem by itself, but it does mean that conclusions cannot be drawn beyond them. The authors did add a few (admittedly, good) sentences to the Discussion acknowledging this issue, but kept the overall framing of the paper intact, including, for instance, claiming “a bimodal distribution (...) at global scales” in the Abstract (lines 293-294). Even though there exist some data points on all continents, (even the suggestion of) global extrapolation is not justified, in my opinion.

Regarding my major comment 2, I thank the authors for their response, but it is disappointing that they decided not to implement any revisions related to this, in my view, important comment.

To illustrate my point, let us consider a hypothetical situation in which the “level of deciduousness” is completely random. This could be represented by a collection of 1000 random numbers between 0 and 1, each number indicating the fraction of the year in which a certain individual sheds its leaves. This distribution will not be bimodal, as can be seen here:

Now let us classify these individuals as deciduous versus evergreen based on whether they shed leaves some part of the year. We can set an arbitrary cutoff at 0.05; so, at a “level of deciduousness” above this value (5% of the year or more), an individual is called deciduous and will be assigned value 1. If at most 5% of the year the individual sheds its leaves, it is called evergreen and will be assigned value 0. If we then plot our frequency distribution of deciduousness, we would end up with the following:

Thus, starting with a completely random distribution of deciduousness, we ended up with a “bimodal” distribution for the sole reason that we defined deciduousness as “having some level of deciduousness” (more than 5%, to be exact). In other words, our inference of bimodality would be erroneous. Hence my request to test for actual bimodality in deciduousness as a continuous variable without a priori classifying the two types of trees as being different in a similar way as in my hypothetical example.

The plots for growing season length for 400 species in North America seem like a step in the right direction, but this is insufficiently substantiated and not included from the revised paper.

The statement in the response that “*Unlike temperate and boreal regions, where leaf shedding and retention are genetically determined, tropical regions present a scenario where deciduousness is more directly shaped by external environmental factors.*” is interesting, but it indicates that the only areas where a true test for bimodality would be possible are those where data are lacking.

Reviewer #4:

Remarks to the Author:

In their revised version, the authors addressed the main technical concerns of reviewer 3 and 4. The presentation of the predictive performance is now much more transparent, e.g. by including the scatter plots to compare predicted and observed values, as well as the comparison to a purely spatial model. Also, I acknowledge the restriction to the northern hemisphere. Looking at the very few data points outside Europe and the US, I'm still not convinced about the quality of predictions in e.g. Siberia. Even though the authors could show that extrapolation in feature space is rather low (S9a), other factors which were not included in the model may lead to different patterns. Since hardly any data are available from outside USA and Europe, I would still be very careful with spatial extrapolation to these areas. Still, the risk is considerably minimized compared to the previous version of the manuscript where the model was used to make global predictions.

My other remaining concern is the one on the reference data and to which degree they are representative for natural patterns. Even though the authors are well aware of the issue and minimized it as best as possible, the issue remains. I'm not sure that the data are suitable to answer the research questions of this study but since this topic is beyond my field of expertise, I have to leave this question to other reviewers.

Minor issues:

Color in S9a: I understand that the authors want to highlight that all values are located towards the end of the scale, but it does not allow for a differentiation. An other color scale should be used that allows for a better differentiation in the range of the values.

"global south" (socioeconomic classification) should be replaced by "Southern Hemisphere" when explaining the limitations of the prediction model.

RESPONSE TO REVIEWER 1

[Comment 1]

First of all, I thank the authors for their responses. In my previous review, I raised two major points and several minor points. Although the minor points have been implemented to my satisfaction, I am sorry to say that the authors did not treat my major points as such.

Response 1: We thank the reviewer for the positive and constructive feedback on our paper. We also very much appreciate the concerns raised by the reviewer and believe we were able to address all of them as outlined below.

[Comment 2]

My first major comment referred to the fact that the analyses have limited geographic scope. Data that were analysed are mostly restricted to temperate North America and Europe, which, as I explained, is not a problem by itself, but it does mean that conclusions cannot be drawn beyond them. The authors did add a few (admittedly, good) sentences to the Discussion acknowledging this issue, but kept the overall framing of the paper intact, including, for instance, claiming “a bimodal distribution (...) at global scales” in the Abstract (lines 293-294). Even though there exist some data points on all continents, (even the suggestion of) global extrapolation is not justified, in my opinion.

Response 2: We thank the reviewer for this comment. We have rephrased the word “global” to “Hemisphere-wide” in the Abstract and in the main text wherever necessary. Additionally, we have expanded the discussion regarding extrapolating results based on our dataset, in which the majority of data come from the US and Europe, in lines 508-513.

[Comment 3]

Regarding my major comment 2, I thank the authors for their response, but it is disappointing that they decided not to implement any revisions related to this, in my view, important comment. To illustrate my point, let us consider a hypothetical situation in which the “level of deciduousness” is completely random. This could be represented by a collection of 1000 random numbers between 0 and 1, each number indicating the fraction of the year in which a certain individual sheds its leaves. This distribution will not be bimodal, as can be seen here:

Now let us classify these individuals as deciduous versus evergreen based on whether they shed leaves some part of the year. We can set an arbitrary cutoff at 0.05; so, at a “level of deciduousness” above this value (5% of the year or more), an individual is called deciduous and will be assigned value 1. If at most 5% of the year the individual sheds its leaves, it is called evergreen and will be assigned value 0. If we then plot our frequency distribution of deciduousness, we would end up with the following:

Thus, starting with a completely random distribution of deciduousness, we ended up with a “bimodal” distribution for the sole reason that we defined deciduousness as “having some level of deciduousness” (more than 5%, to be exact). In other words, our inference of bimodality would be erroneous. Hence my request to test for actual bimodality in deciduousness as a continuous variable without a priori classifying the two types of trees as being different in a similar way as in my hypothetical example.

The plots for growing season length for 400 species in North America seem like a step in the right direction, but this is insufficiently substantiated and not included from the revised paper.

The statement in the response that “Unlike temperate and boreal regions, where leaf shedding and retention are genetically determined, tropical regions present a scenario where deciduousness is more directly shaped by external environmental factors.” is interesting, but it indicates that the only areas where a true test for bimodality would be possible are those where data are lacking.

Response 3: We thank the reviewer for the follow-up comment on our treatment of leaf phenology within our study. We have to point out that this concern relies on the assumption that the distribution of deciduousness across species is uniform, as the reviewer hypothesizes here. However, the actual distribution of deciduousness is bimodal instead of uniform as we have shown in the previous response. This is now

shown in the paper as Fig. S33. The mean growing season length of deciduous trees is between 150 and 250 days, whereas all evergreen trees per definition have a growing season length of 365 days.

We also want to emphasize that we are interested in bimodality across plots in our study, not within plots. In simulation 1 of Figure 3, we show that it is definitely possible to obtain a non-bimodal distribution even if, at the species-level, deciduousness is defined in a binary way. It is not true that our definition of deciduousness automatically leads to bimodality.

Nevertheless, we now added a paragraph on our categorical classification of deciduousness to the caveat part of the discussion (lines 503-507).

Fig. S33 | Bimodal patterns in deciduousness when treating leaf shedding as a continuous trait. To demonstrate that the actual underlying distribution of leaf shedding, or deciduousness, is bimodal, we used species-level data on growing season length from Zohner et al. (2017), which includes data for more than 400 species from North America, Europe and Asia. This figure shows that across all continents, there is a clear bimodal distribution of evergreen and deciduous species, which was also statistically confirmed ($P < 0.05$) by a Hartigan's dip test.

RESPONSE TO REVIEWER 4

[Comment 1]

In their revised version, the authors addressed the main technical concerns of reviewer 3 and 4. The presentation of the predictive performance is now much more transparent, e.g. by including the scatter plots to compare predicted and observed values, as well as the comparison to a purely spatial model. Also, I acknowledge the restriction to the northern hemisphere. Looking at the very few data points outside Europe and the US, I'm still not convinced about the quality of predictions in e.g. Siberia. Even though the authors could show that extrapolation in feature space is rather low (S9a), other factors which were not included in the model may lead to different patterns. Since hardly any

data are available from outside USA and Europe, I would still be very careful with spatial extrapolation to these areas. Still, the risk is considerably minimized compared to the previous version of the manuscript where the model was used to make global predictions.

Response 4: We thank the reviewer for this constructive and positive comment. We have rephrased the word “global” to “Hemisphere-wide” in the Abstract and in the main text wherever necessary. Additionally, we have expanded the discussion regarding extrapolating results based on a biased dataset that contains more data on the US and Europe in lines 508-513.

[Comment 2]

My other remaining concern is the one on the reference data and to which degree they are representative for natural patterns. Even though the authors are well aware of the issue and minimized it as best as possible, the issue remains. I’m not sure that the data are suitable to answer the research questions of this study but since this topic is beyond my field of expertise, I have to leave this question to other reviewers.

Response 5: We have expanded the discussion of human impacts in lines 498-502.

[Comment 3]

Color in S9a: I understand that the authors want to highlight that all values are located towards the end of the scale, but it does not allow for a differentiation. An other color scale should be used that allows for a better differentiation in the range of the values.

Response 6: Changed accordingly.

[Comment 4]

“global south” (socioeconomic classification) should be replaced by “Southern Hemisphere” when explaining the limitations of the prediction model.

Response 7: Changed accordingly in the discussion.

[Comment 5]

I briefly reviewed the code. The study is, in it's current form , not reproducible because the data to run the scripts are missing (data folders and model output folders are empty). Also the structure can be improved or a documentation is required describing the order of the scripts. Currently, one can only guess by file name.

Also note that the code is using the raster package which is deprecated, i.e. the code cannot be used anymore in the future. Transition to stars is required to make the code reusable.

Response 8: We have deposited the source data in Zenodo. In addition, we replaced the use of “raster” package by the “terra” package.